# Simultaneous in-situ measurements of small-scale structures in neutral, plasma, and atomic oxygen densities during WADIS sounding rocket project

Boris Strelnikov[1], Martin Eberhart[4], Martin Friedrich[5], Jonas Hedin[3], Mikhail Khaplanov[3†],
Gerd Baumgarten[1], Bifford P. Williams[6], Tristan Staszak[1], Heiner Asmus[1], Irina Strelnikova[1],
Ralph Latteck[1], Mykhaylo Grygalashvyly[1], Franz-Josef Lübken[1], Josef Höffner[1], Raimund Wörl[1],
Jörg Gumbel[3], Stefan Löhle[4], Stefanos Fasoulas[4], Markus Rapp[2], Aroh Barjatya[7], Michael J. Taylor[8],
and Pierre-Dominique Pautet[8]

[1]Leibniz-Institute of Atmospheric Physics at the Rostock University, Kühlungsborn, Germany
[2]Deutsches Zentrum für Luft- und Raumfahrt, Institut für Physik der Atmosphäre, Oberpfaffenhofen, Germany
[3]Department of Meteorology (MISU), Stockholm University, Stockholm, Sweden
[4]University of Stuttgart, Institute of Space Systems, Stuttgart, Germany
[5]Graz University of Technology, Graz, Austria
[6]GATS, Boulder, USA
[7]Embry-Riddle Aeronautical University, FL, USA
[8]Center for Atmospheric and Space Sciences, Utah State University, Logan, Utah, USA
[†]Deceased

**Correspondence:** B. Strelnikov (strelnikov@iap-kborn.de)

**Abstract.** In this paper we present an overview of measurements conducted during the WADIS-2 rocket campaign. We investigate the effect of small-scale processes like gravity waves and turbulence on the distribution of atomic oxygen and other species in the MLT region. Our analysis suggests that density fluctuations of atomic oxygen are coupled to fluctuations of other constituents, i.e., plasma and neutrals. Our measurements show that all measured quantities, including winds, densities, and temperatures, reveal signatures of both waves and turbulence. We show observations of gravity wave saturation and breakdown together with simultaneous measurements of generated turbulence. Atomic oxygen inside turbulence layers shows two different spectral behaviors, which might imply change of its diffusion properties.

## 1  Introduction

The mesosphere, lower thermosphere (MLT) region is host of phenomena that are connected to dynamic and chemical processes which are still not fully understood. Thus, e.g., it is generally accepted that atmospheric gravity waves (GW) play an essential role in the dynamics of this region and that they couple it with the lower and upper atmosphere (e.g., Becker and Schmitz, 2002; Fritts and Alexander, 2003; Alexander et al., 2010). When propagating, GW might dissipate and thereby generate turbulence (e.g., Yamada et al., 2001; Selvaraj et al., and references therein). Apart of the momentum deposition, which is a key coupling process, this also affects mixing and redistribution of trace constituents (e.g., Fukao et al., 1994; Bishop et al., 2004). One of the most important trace constituents in MLT is atomic oxygen (O) which plays an essential role in the chemistry and

energy budget of the mesopause region (e.g., Mlynczak and Solomon, 1993). It is the major reactive trace constituent in the mesosphere/lower thermosphere (MLT) region and it plays a crucial role in different chemical reactions involved in airglow excitation or ion chemistry (e.g., Walterscheid et al., 1987; McDade et al., 1986; McDade, 1998; Marsh et al., 2006; Caridade et al., 2013; Lednyts'kyy et al., 2015). The lifetime of atomic oxygen varies with altitude from seconds at ∼50 km to months at ∼100 km (e.g., Torr, 1985; Smith et al., 2010).

Chemical heat released during exothermic reactions involving atomic oxygen is one of the main contributors to the energy budget of this region (e.g., Mlynczak and Solomon, 1991; Mlynczak and Solomon, 1993). These reactions yield chemical heating rates in the mesopause region of several K/day which is comparable (or even competitive) to those of turbulent heating (Lübken, 1997; Lübken et al., 2002) as well as direct heating due to solar radiation (e.g., Fomichev et al., 2004; Feofilov and Kutepov, 2012; Lübken et al., 2013, and references therein). Oxygen is involved in exothermic reactions with hydroxyl (OH) which emission bands have been extensively used to study the mesopause-region temperature, gravity waves, and tides (e.g., Hines and Tarasick, 1987; Taylor et al., 1995, 1997, 2009; Snively, 2013; Fritts et al., 2014; Egito et al., 2017, and references therein).

It is believed that at altitudes where the O-lifetime is very long, i.e., roughly above 90 km, variations in O-density mainly result from dynamical processes such as GW, tides, and the large scale circulation. Whereas at lower heights chemical processes may play a crucial role in forming O-density variations. Such generalized conditions, however, can considerably vary from individual cases and are not suitable for case studies. It is also known, that mesopause region is very active dynamically. It is region where GW break and a persistent turbulence field plays crucial role in global circulation (e.g., Becker and Schmitz, 2002; Fritts and Alexander, 2003; Rapp et al., 2004). However, it is not known, for instance, how turbulence influences the O-density distribution, its diffusion properties and whether it can affects chemical heat release e.g., by changing reaction rates.

This implies that in order to properly characterize the chemical and dynamical state of the mesopause region, it is important to know the altitude resolved concentration of atomic oxygen alongside with the state of the background atmosphere including its thermal structure and dynamical parameters. Also, a suitable justification of chemical life time of atomic oxygen as well as characteristic times for dynamical processes have to be assessed for proper interpretation of measurements.

All these aspects stress the importance of common-volume measurements of the oxygen density together with temperature and density of the background neutral atmosphere with a sufficiently high altitude resolution.

This paper aims at two things. First, it is to provide an overview of the WADIS-2 sounding rocket campaign and measured parameters, and second, to demonstrate that gravity wave motions and turbulence effects distribution of atomic oxygen in the nighttime MLT region. We present results of in-situ and some ground based measurements obtained in the frame of the WADIS sounding rocket mission. We introduce new high resolution O-density measurements in connection with other parameters of the atmosphere. The paper is structured as follows. In Sec. 2 we briefly describe the WADIS-2 mission and measurement techniques used in this study. In Sec. 3 we present measurement results followed by a more detailed analysis in Sec. 4. In Sec. 5 we critically discuss our findings and finally, in Sec. 6 we summarize main results and give an outlook to our next rocket borne measurements.

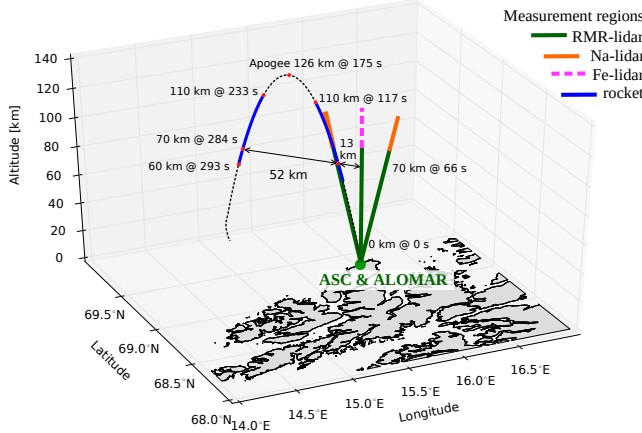

**Figure 1.** Schematics of WADIS-2 rocket experiment. Solid green and orange, and dashed magenta lines show ALOMAR RMR-, Na-, and IAP Fe-lidar measurement volumes, respectively. Black dashed line shows rocket trajectory. Blue lines show rocket measurement range used in this study.

## 2 WADIS-2 rocket campaign and instrumentation

The WADIS sounding rocket mission led by the Leibniz-Institute of Atmospheric Physics (IAP) in Kühlungsborn, Germany, in partnership with the Institute of Space Systems (IRS) in Stuttgart and contributions from Austria, Sweden, the USA, and Norway, comprised two field campaigns conducted at the Andøya Space Center (ACS) in northern Norway (69°N, 16°E).

The first campaign was conducted in June 2013 and the second in March 2015. WADIS stands for "Wave propagation and dissipation in the middle atmosphere: Energy budget and distribution of trace constituents". The mission aimed at studying the propagation and dissipation of gravity waves (GW) and measuring concentration of atomic oxygen simultaneously. Such measurements allow for estimation of the contribution of chemical and turbulent heating to the energy budget of the MLT, as well as the transport of atomic oxygen by waves and turbulence. For a more detailed mission description the reader is referred

to Strelnikov et al. (2017). The Arctic Lidar Observatory for Middle Atmosphere Research (ALOMAR, von Zahn et al., 1995) is located close to the launch site and was an integral part of the entire WADIS mission.

The launch window for the second sounding rocket campaign was scheduled around local midnight to ensure full night-time conditions. A large number of ground-based optical instruments were supporting the WADIS-2 rocket campaign. The ALOMAR Rayleigh/Mie/Raman (RMR)-, Na Weber, and IAP Fe-lidar were running continuously throughout the campaign

period whenever weather permitted. All three lidars measure temperatures profiles along the beam direction as shown in Fig. 1. The instruments and temperature retrieval techniques for these lidars are described elsewhere (von Zahn et al., 2000; Hauchecorne and Chanin, 1980; She et al.; Arnold and She, 2003; Höffner and Fricke-Begemann, 2005; Lautenbach and Höffner, 2004; Höffner and Lautenbach, 2009). Additionally, RMR- and Na-lidars measure line-of-sight wind speed in the altitude ranges 20–80 and 80–110 km, respectively (Baumgarten, 2010; Arnold and She, 2003).

The Advanced Mesospheric Temperature Mapper (AMTM) by Utah State University (Pautet et al., 2014) was observing the night glow emissions at 1523.68 and 1542.79 nm. These observations yield measurements of horizontal temperature field (see Fig. 4 and Wörl et al., 2019, for a detailed discussion of measurements during WADIS-2 campaign).

The Middle Atmosphere ALOMAR Radar System, MAARSY, (Rapp et al., 2011; Latteck et al., 2012) operated by IAP located close to the rocket launch site was continuously running. MAARSY was used to detect polar mesospheric winter echoes (PMWE) in case if they occur (Latteck and Strelnikova, 2015).

The instrumented WADIS-2 payload was almost identical to that one launched during the first campaign (see Strelnikov et al., 2017, for details), except that the instruments were tuned for the polar night launch conditions.

The front and rear decks of the WADIS payloads were equipped with identical CONE ionization gauges to measure turbulence, neutral air density, and temperature on up- and downleg (Strelnikov et al., 2013). CONE measures density of neutral air with altitude resolution of $\sim$30 cm. Making use of laboratory calibrations allows to derive absolute density altitude-profile. The measured density profile, in turn, can be integrated assuming hydrostatic equilibrium to yield temperature profile (see e.g., Strelnikov et al., 2013, for details). High sensitivity of the CONE instrument, which allows to resolve density fluctuations of $\sim$0.01 %, makes it possible to derive turbulent parameters by analyzing the spectra of these fluctuations. The derivation technique of turbulent parameters is described in detail elsewhere (Lübken, 1992; Lübken et al., 1993; Lübken, 1997; Strelnikov et al., 2003; Strelnikov et al., 2013). Briefly, a theoretical spectral model of turbulence is fitted to a Fourier or wavelet spectrum of the measured relative density fluctuations, which are shown to be a conservative and passive turbulence tracer in MLT (Lübken, 1992; Lübken et al., 1993; Lübken, 1997). The key-feature of this technique is that the theoretical model must reproduce spectrum of turbulent tracer (scalar) in both inertial (i.e. $\propto k^{-5/3}$) and viscous (or dissipation) subranges. Transition between these subranges takes place at the so called inner scale, $l_0$, which is related to the turbulence energy dissipation rate, $\varepsilon$ as $l_0 = C(\nu^3/\varepsilon)^{(1/4)}$, where $\nu$ is the kinematic viscosity and the constant $C$ is of the order 10. Also, Lübken et al. (1993) and Lübken (1997) showed that different spectral models, in particular those by Heisenberg (1948) and by Tatarskii (1971), yield close results of the energy dissipation rates.

A positive ion probe (PIP) operated by the University of Technology in Graz (TUG), Austria, and novel Langmuir probe (LP) developed and operated by Embry-Riddle Aeronautical University in Florida, USA, yielded high-resolution positive ion and electron density measurement, respectively. Both these plasma probes were mounted on booms located on the rear deck of the payload. Both WADIS payloads also carried the wave propagation experiment operated by TUG for precise measurements of absolute electron densities (Friedrich, 2016).

Two instruments, the FIPEX and an airglow photometer, were used to measure atomic oxygen densities. These instruments, utilized entirely different measurement techniques. Photometers yield precise absolute density measurements, whereas FIPEX yields high altitude resolution data. The absolute values of the FIPEX-measurements were validated by the Photometers (see companion paper by Eberhart et al., 2018, for more details).

The FIPEX instruments developed by IRS, yield profiles of atomic oxygen densities with high altitude resolution of $\sim$20 m (see Eberhart et al., 2015). Photometers operated by the Meteorological Institute at Stockholm University (MISU) measured

oxygen densities using a well established reliable technique applied before on a large number of sounding rockets (e.g., Hedin et al., 2009).

FIPEX stands for „Flux-Probe-Experiment", it employs solid electrolyte sensors having gold electrodes with selective sensitivity towards atomic oxygen. A low voltage is applied between anode and cathode pumping oxygen ions through the electrolyte ceramic (yttria stabilized zirconia, YSZ). The current measured is proportional to the oxygen flux. A detailed description of measurements conducted by FIPEX during WADIS mission is provided by Eberhart et al. (2015) and Eberhart et al. (2018) for the first and second campaign, respectively.

The MISU airglow photometer measures emission of the molecular oxygen Atmospheric Band around 762 nm from the overhead column, from which volume emission rate is inferred by differentiation. A theory and application of oxygen density retrievals from the Atmospheric Band emissions is discussed in details in the companion paper by Grygalashvyly et al. (2019).

The second MISU photometer on the payload measured the emission from the (0-0) band of the $N_2^+$ 1st negative band system centered at 391.4 nm. This emission is a sign of precipitating auroral electrons and thus a sensitive indicator of auroral activity.

The WADIS-2 sounding rocket was launched on 5th of March 2015 at 01:44:00 UTC, that is during full night-time conditions. Fig. 1 shows the geometry of the WADIS-2 experiment. The black dashed parabola shows the actual rocket trajectory. The blue profiles show parts of the rocket trajectory which yielded measurements used in this paper. The solid green, orange, and dashed magenta lines show direction and altitude range for RMR-, Na, and Fe-lidar, respectively. It is seen, that the north-west (NW) directed lidar beam was co-located with the ascending part of the rocket trajectory.

## 3    Data

In this section we show data obtained by ground-based instruments around the rocket launch time and some profiles measured on board the WADIS-2 sounding rocket. We start with measurements of the background state of the atmosphere as observed during the night of the rocket launch. Then we compare upleg and downleg density measurements conducted by different instruments. These profiles are then used in Sec. 4 for a detailed fluctuation analysis. Finally, we demonstrate that the high-resolution FIPEX-measurements yielded new, geophysically meaningful data.

### 3.1    Background

The background state of the atmosphere was continuously monitored by the ALOMAR lidars and radars. MAARSY did not observe any echoes during the night of the rocket launch. That is, the WADIS-2 rocket launch was under conditions of confirmed absence of polar mesospheric winter echoes, PMWE. Fig. 2 shows volume reflectivity measured by MAARSY during 5 of March, i.e. the day of the rocket launch. Some short-living echoes were observed around noon and in the late evening, but not in the morning when WADIS-2 rocket was launched. We recall here that this sounding rocket mission did not aim at studying PMWE, so that the presence of PMWE was not a criterion for rocket launch.

Thanks to favorable weather conditions, the lidars were able to measure both temperature and wind fields for several hours around rocket launch time.

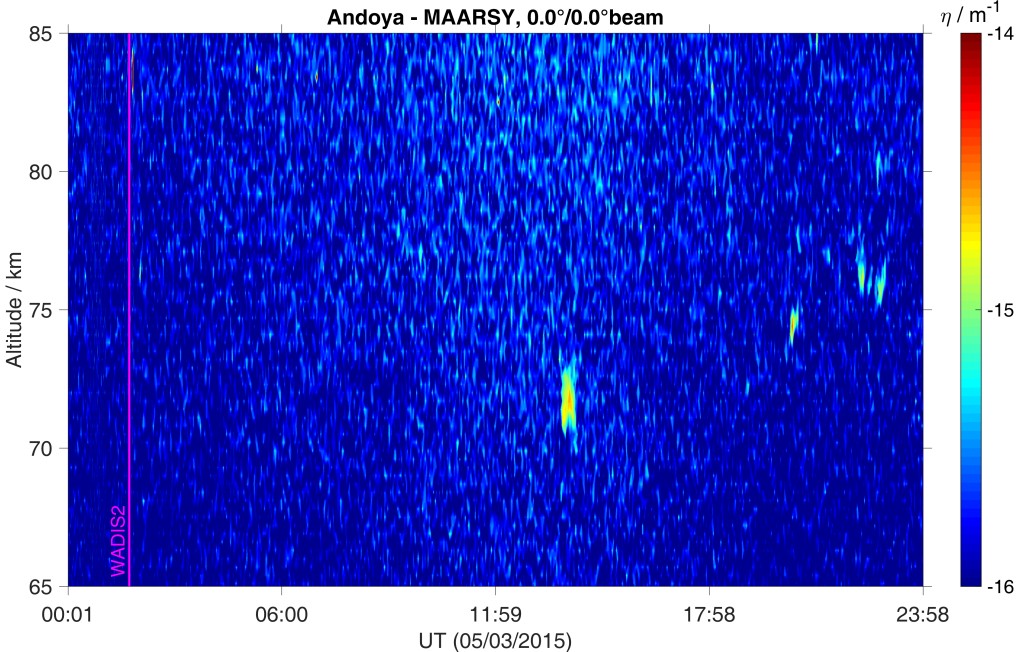

**Figure 2.** Volume reflectivity measured by MAARSY on 5 of March. Some short-living echoes were observed around noon and in the late evening but not around the WADIS-2 rocket launch.

Fig. 3 shows the temperature field measured by the IAP RMR- and Fe-lidars from 20 up to ∼100 km altitude around the time of the WADIS-2 rocket launch. Fig. 3 utilizes measurements by the vertical beam of RMR-lidar since the mobile Fe-lidar only measures vertically. Also, the seeding temperature for derivation of RMR-temperatures was taken from Fe-lidar measurements. Signatures of long period waves are clearly seen above ∼65 km altitude in both RMR- and Fe-lidar measurements.

Horizontal temperature field observed by the AMTM (Pautet et al., 2014) shown in Fig. 4 also reveals clear large scale structures around the WADIS-2 launch time. Wörl et al. (2019) analyzed the co-located temperature measurements by both Fe-lidar and AMTM in detail and concluded that the most pronounced wave signatures reveal periods of 24, 12, and 8 hours i.e., they are most probably created by tides.

Fig. 5 shows measurements by the Na-lidar conducted throughout the night of the WADIS-2 launch. Similar to Fe-temperature

these measurements show prominent signatures of long period waves above 80 km altitude in both temperature and wind fields.

All these data contain also smaller scale fluctuations which result from gravity waves and turbulence. To see such small-scale fluctuations better and analyze them properly, one has to subtract the large-scale background (including tides) from the measurements shown in Fig. 3, 4, and 5 (see e.g., Strelnikova et al., 2019). In the next section we focus on small-scale fluctuations of different quantities. We analyze rocket-borne instant measurements and further compare these with profiles

measured by the ground-based instruments at the time of rocket launch.

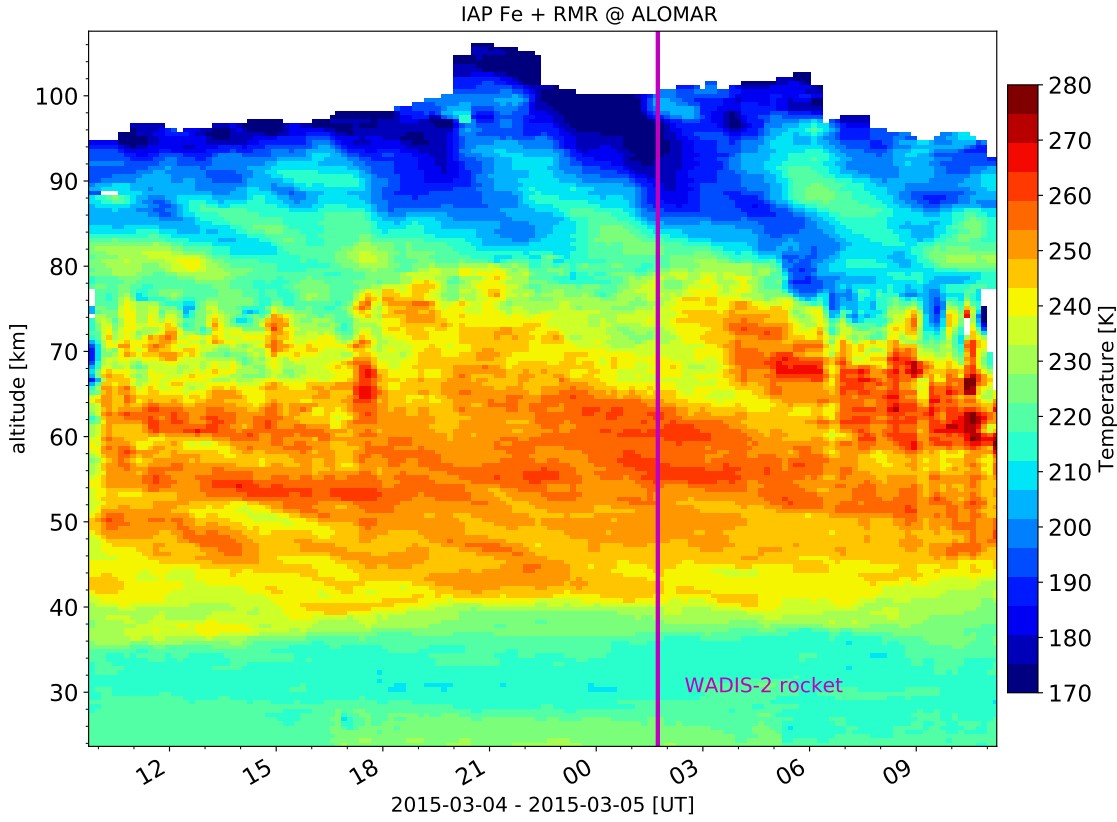

**Figure 3.** Combined RMR- and Fe-lidar temperature measurements during the night of the WADIS-2 rocket launch, i.e. 4 to 5 of March 2015.

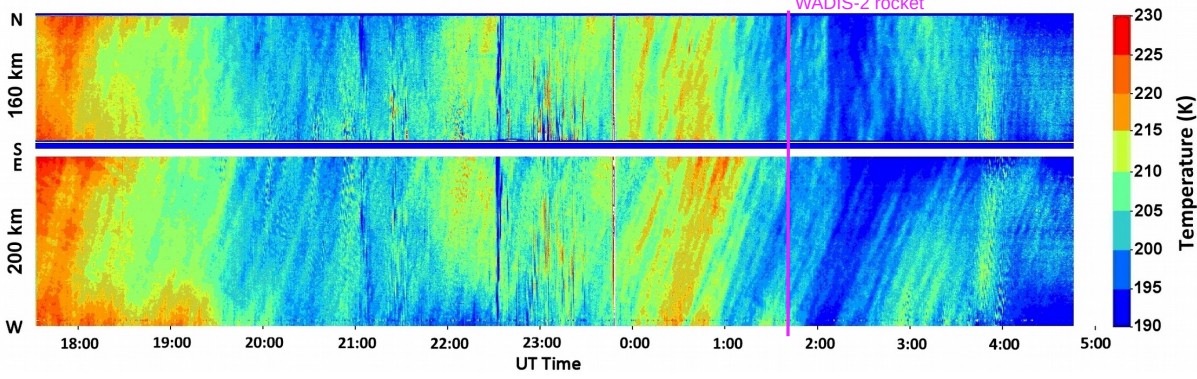

**Figure 4.** NS and WE keogram summary of the AMTM temperature measurements obtained during the night of 4 to 5 March 2015.

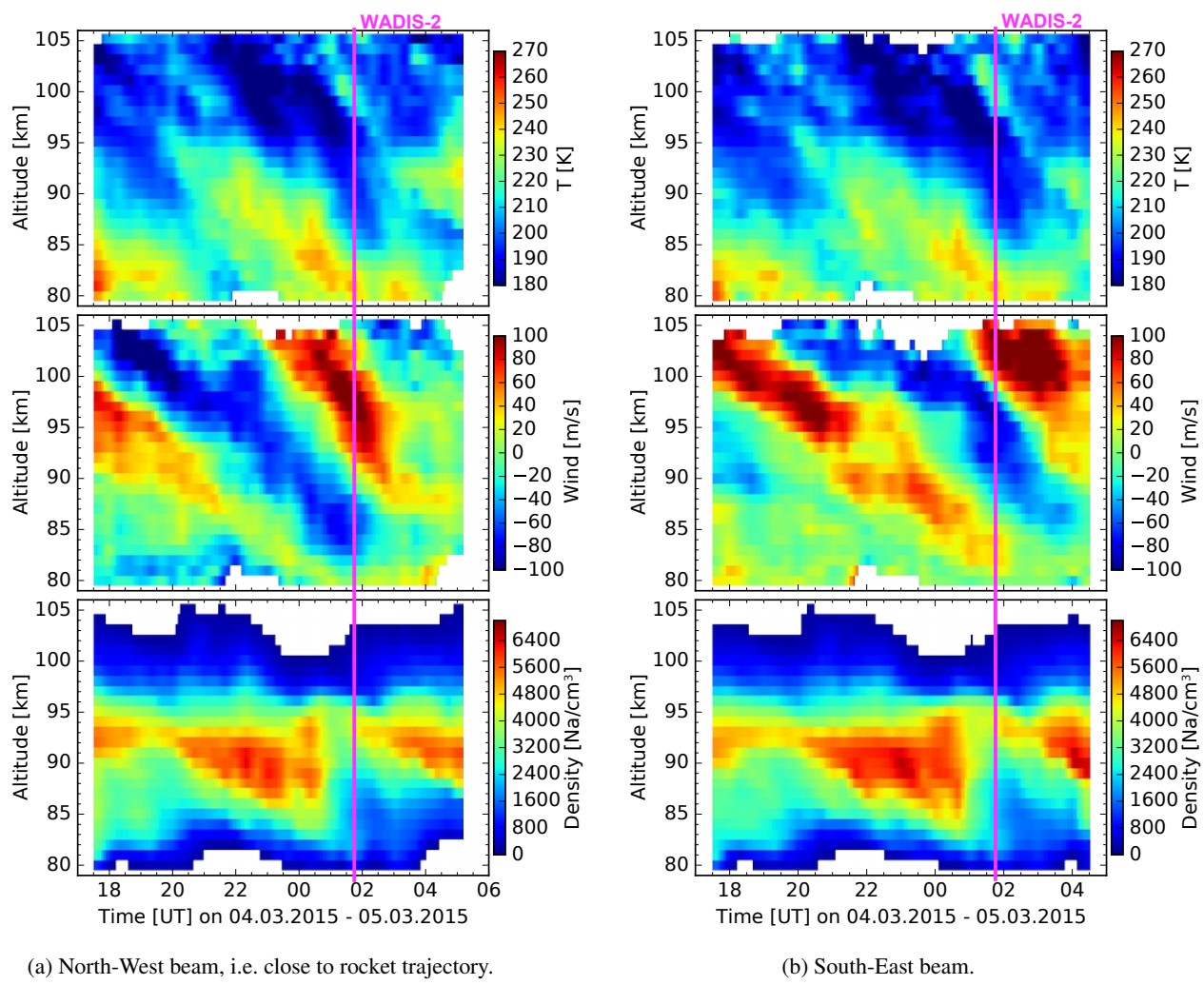

(a) North-West beam, i.e. close to rocket trajectory.

(b) South-East beam.

**Figure 5.** Na-lidar measurements during the night of the WADIS-2 rocket launch, i.e. 4 to 5 of March 2015. Rocket was launched at 1:44 UT

## 3.2 In-situ measurements

The WADIS-2 rocket was launched at 1:44 UTC and reached an apogee of 126 km. The measurement phase started at about 60 km altitude after nose-cone and motor separation. However the best quality data was obtained above 70 km height, mostly due to favorable aerodynamic conditions.

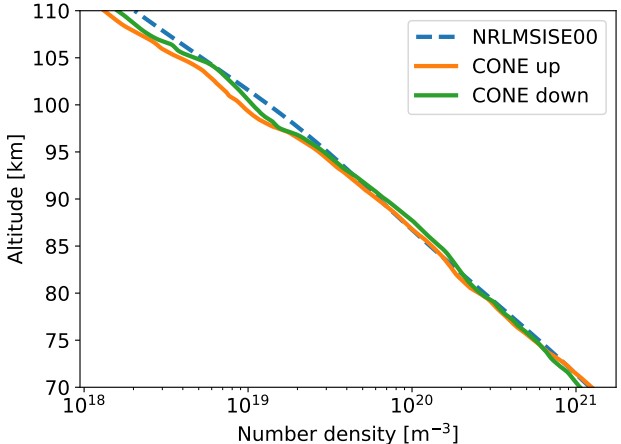

**Figure 6.** Rocket-borne density measurements by the ionization gauge CONE for upleg and downleg (orange and green lines, respectively). Blue dashed line shows the NRL-MSISE00 climatology for the time of WADIS-2 launch, i.e. 5 of March 2015, 01:44:00 UTC.

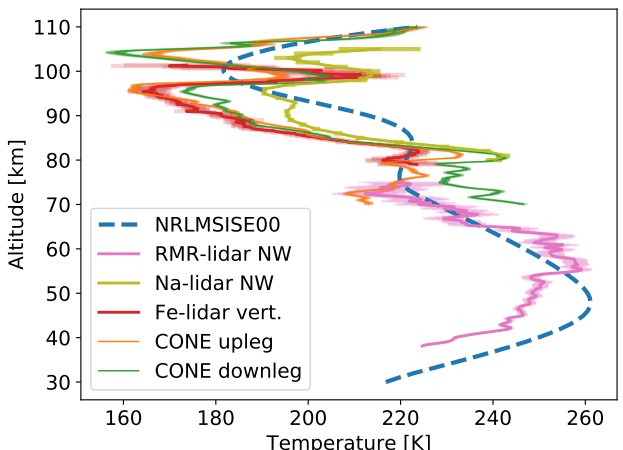

**Figure 7.** Temperatures derived from the densities shown in Fig. 6 assuming hydrostatic equilibrium: orange and green for upleg and downleg, respectively. Profiles in magenta, yellow, and red show measurements by RMR-, Na-, and Fe-lidars, respectively. Blue dashed line shows NRLMSISE-00 reference atmosphere for the time of WADIS-2 launch (5 of March 2015, 01:44:00 UTC).

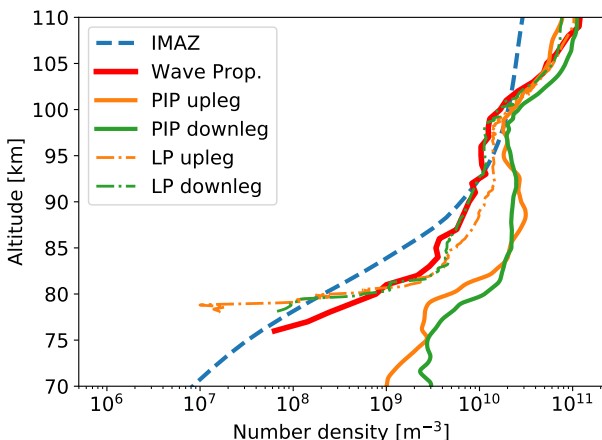

**Figure 8.** Densities of positive ion (solid orange and green lines) and electron (dash-dotted orange and green profiles) densities measured by the PIP and LP instruments, respectively. Bold solid red line shows results of the wave propagation experiment. Blue dashed line shows electron density from the empirical ionospheric model for the auroral zone, IMAZ, derived for the time of the WADIS-2 flight (see text for details).

Figs. 6 and 7 show profiles of neutral air densities and temperatures measured in-situ by the CONE instrument. Orange and green lines show up- and downleg measurements, respectively. Blue dashed profiles represent NRLMSISE-00 reference atmosphere (Picone et al., 2002). The spin frequency of WADIS-2 rocket of 3.27 Hz which modulated the raw data was filtered out by applying a notch filter. Additionally, the shown in-situ measured densities were smoothed by running average of a length

of ∼200 m.

Both up- and downleg profiles look very similar in terms of mean values and oscillations. The background atmosphere reveals a typical winter state (see e.g., Strelnikov et al., 2013, where a collection of rocket-borne measurements for different seasons is shown). Also, the observed turbulence activity shown in Sec. 4 demonstrates a typical for winter behavior. The temperature profiles clearly show some GW-signatures with amplitudes of up to 15 K at altitudes below 80 km. The height

range between ∼83 and 90 km reveals very low GW amplitudes, i.e. temperature fluctuations of 1 K and less. A temperature increase of ∼40 K reminiscent of mesospheric inversion layers (MIL) similar to those analyzed by Szewczyk et al. (2013) is seen between 95 and 100 km and is discussed in Sec. 4. Some small-scale GWs with amplitudes between 1 and 5 K and vertical wavelength of the order kilometer are superposed on this large temperature disturbance. Fig. 7 additionally shows temperature profiles in magenta, yellow, and red measured by RMR-, Na-, and Fe-lidars, respectively.

The neutral density profiles in Fig. 6 also show some oscillations that can be attributed to gravity waves, which will be analyzed in detail in Sec. 4.

In Fig. 8 we show profiles of electron and positive ion densities measured in-situ on both up- and downleg. Dashed-dotted and solid profiles show electron and positive ion data, respectively. The bold solid red line shows measurements results of the

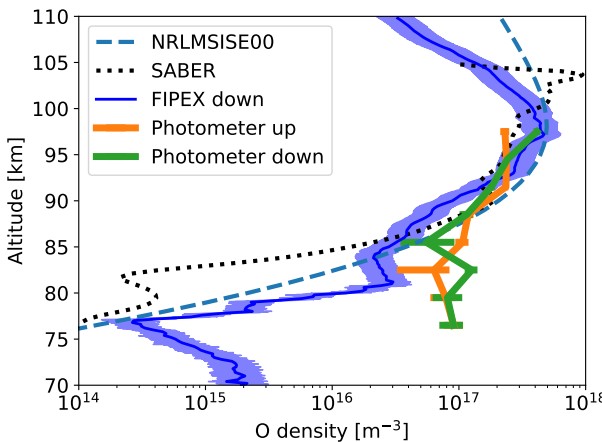

**Figure 9.** Atomic oxygen densities measurements. Bold orange and green lines with error-bars show photometer measurements on up- and downleg, respectively. FIPEX downleg data is shown by blue profile with shaded area showing measurement errors. Black dotted line shows SABER measurements (Level 2A, O event 20 orbit 71729). Blue dashed line shows NRLMSISE-00 model data for the time of rocket launch.

wave propagation experiment. The two ion density profiles are also quite similar and also reveal some wave signatures. The two electron density profiles also demonstrate that ionospheric background at the rocket up- and downleg is rather similar. Comparison with the electron density profile from the empirical ionospheric model for the auroral zone, IMAZ (McKinnell and Friedrich, 2007) shown in Fig. 8 (blue dashed line) shows that ionization level of the ionosphere was moderately high. This is in accord with the fact that some aurora was seen throughout the night of these observations. The inputs for IMAZ model are F10.7 solar flux index of 137.9 Jy (1 Jy = $10^{-26}$ W m$^{-2}$Hz$^{-1}$), the planetary magnetic Ap index 5, and riometer absorption @ 27.6 MHz of 0.076 dB. Activity indices, the solar F10.7 and Ap index, were obtained from the GSFC/SPDF OMNIWeb interface at https://omniweb.gsfc.nasa.gov. The integral riometer absorption was estimated from the electron density measurements by wave propagation experiment based on Friedrich and Torkar (1983).

The overhead emission seen by the second MISU photometer was varying (both increasing and decreasing at times) during the flight indicating that the auroral emission was variable in time. The auroral emission was relatively weak with peak total band radiances of 700–800 Rayleighs (around $6 \cdot 10^7$ photon $\cdot$ s$^{-1}$ str$^{-1}$ cm$^{-2}$).

The PIP and LP instruments yield measurements of relative densities of positive ions and electrons, respectively. The wave propagation experiment yields accurate measurements of absolute electron densities, which are used to normalize the PIP- and LP-measurements. The normalization was made at an altitude of ∼115 km, where quasi-neutrality condition is well satisfied for ionospheric plasma (see e.g., Friedrich, 2016; Asmus et al., 2017, for more details). The large difference between positive ion and electron densities at altitudes ∼80–95 km was studied in detail by Asmus et al. (2017) and was shown to be due to charged dust particles.

Fig. 9 shows atomic oxygen density profiles measured by the photometer on up- and downleg in orange and green, respectively. The FIPEX measurements are only shown for the descending part of the WADIS-2 rocket flight, because we are mostly confident in their absolute values and will use this data for further analysis. The blue dashed line shows NRLMSIS-00 data and black dotted line shows SABER retrievals (orbit 71729, event 20) at ∼230 km distant location and ∼4 h before rocket launch. That is, the SABER measurements were not collocated and not simultaneous with the rocket flight and are only shown for qualitative comparison. However, the apparent density increase in SABER data above 100 km must rather be attributed to the observed auroral activity. We recall that the measurement principle of the FIPEX instrument is only sensitive to ambient O-density and does not react on emissions.

We note here that the FIPEX is a new instrument which was first applied for oxygen density measurements on sounding rockets during WADIS-1 rocket campaign (Eberhart et al., 2015). It showed good results and demonstrated principal possibility of a high resolution O-density measurements in MLT. WADIS-2 rocket was additionally equipped with the MISU airglow photometer that indirectly measures O-density from emission of the molecular oxygen Atmospheric Band. Such type of measurements is widely used also from the ground- or satellite-based platforms and is commonly accepted as reliable. The WADIS-2 sounding rocket comprised the both measurement techniques on the same platform which made it possible to closely compare their results. The both WADIS payloads (i.e. for the first and second campaigns) were equipped with several FIPEX sensors which were mounted at different angles relative to rocket velocity (or rocket symmetry) axis. Such multiple configuration aimed at finding the most favorable aerodynamic orientation for the FIPEX sensors. We also note that because of the supersonic rocket velocity the measurement results of most instruments on board sounding rockets require an aerodynamic correction (Gumbel et al., 1999; Gumbel, 2001; Rapp et al., 2001; Hedin et al., 2007; Staszak et al., 2015). By analyzing measurement conditions and comparing the measurement results we chose the best quality FIPEX data for further analysis. For the detailed discussion of all FIPEX measurements the reader is referred to the companion paper by Eberhart et al. (2018).

Fig. 9 also reveals that below ∼85 km the differences between O-densities measured by different techniques are very large. This is a manifestation of disadvantages of the emission methods used for atomic oxygen retrieval. Thus e.g., the technique for deriving O from OH* emission (SABER) is based on assumption of chemical equilibrium for ozone. Several recent works showed that below ∼80–87 km this assumption fails, in particular at high latitudes in March (Fig. 1 in Belikovich et al., 2018; Kulikov et al., 2018). Deviation from chemical equilibrium for ozone results in underestimation of atomic oxygen density (Kulikov et al., 2019). Also the photometric methods of atomic oxygen retrieval via atmospheric band observation (762 nm) are not free of disadvantages. The fitting coefficients for this method were calculated at peak of atmospheric band emission (i.e., above ∼90 km) with an assumption that atomic oxygen recombination is a principle source of $O_2$ ($b^1\Sigma_g^+$). Below this altitude, and in particular around peak of OH* layer, typically located at ∼80-88 km (e.g., Grygalashvyly et al., 2014) , some additional mechanisms of $O_2$ ($b^1\Sigma_g^+$) population may occur (Witt et al., 1979; Kalogerakis, 2019). This leads to overestimation of atomic oxygen calculated by the fit function of McDade et al. (1986).

It is already seen in Fig. 9 that the O-density profiles reveal some oscillations with largest amplitudes below approx. 83 km altitude. It is worth mentioning, that other FIPEX sensors yielded fluctuations data that show the same features (not shown here and discussed in detail in Eberhart et al., 2018)

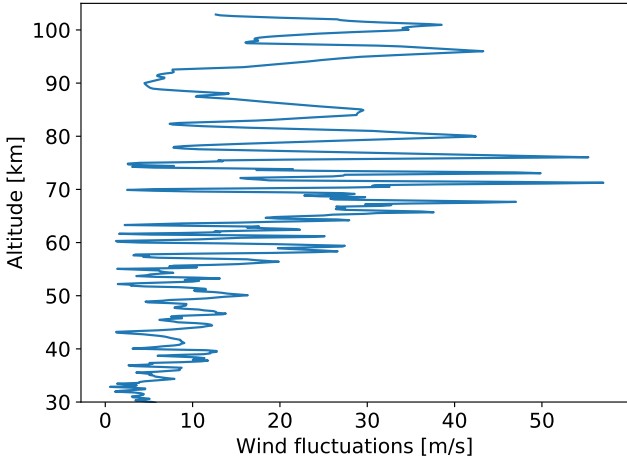

**Figure 10.** Horizontal wind fluctuations (i.e., $\sqrt{u'^2 + v'^2}$, where $u'$ and $v'$ are zonal and meridional wind fluctuations) derived from combined RMR- and Na-lidar measurements for the time of WADIS-2 rocket flight.

The photometer measurements have an effective altitude resolution of 3 km, whereas the catalytic FIPEX sensors exhibit height resolution of ∼20 m. In the next section we examine which advantages the high resolution measurements of the atomic oxygen densities can bring and what the nature of the fluctuations in the FIPEX data is.

## 4 Analysis

To extract small-scale fluctuations from the measured profiles, we subtract the mean background derived as running average over 5 km long vertical window. Similar results can be achieved by e.g., applying a polynomial fit of the same vertical resolution, i.e. 5 km or by means of other techniques (e.g., Strelnikova et al., 2019). We here focus on small vertical length scales <5 km comprising both turbulent fluctuations as well as part of the gravity wave spectrum. We note that choosing a particular cut off wavelength always carries some degree of arbitrariness. For the small-scale structures in MLT and, especially those

produced by turbulence, the background derivation does not affect the data analysis if residual fluctuations data contain all the scales below this limit (i.e. ∼5 km).

Fluctuations of horizontal wind derived from the lidar measurements for the time of the WADIS-2 rocket launch are shown in Fig. 10. It reveals an amplitude increase with height, z, according to the exponential law ($\propto \exp(z/2H)$, where H is scale height). It demonstrates basically, that we observed gravity waves in the entire altitude range from near the ground up to ∼105 km.

Wind fluctuation amplitude increases in the altitude range from 30 up to ∼70 km. Then the amplitude drops within altitude range 70 to 80 km and increases again between 88 and 95 km. This is consistent with the behavior of temperature profiles that show larger wave amplitudes below 80 km altitude, very low amplitudes above that height and large wave amplitude again between 95 and 105 km (Figs. 5 and 7).

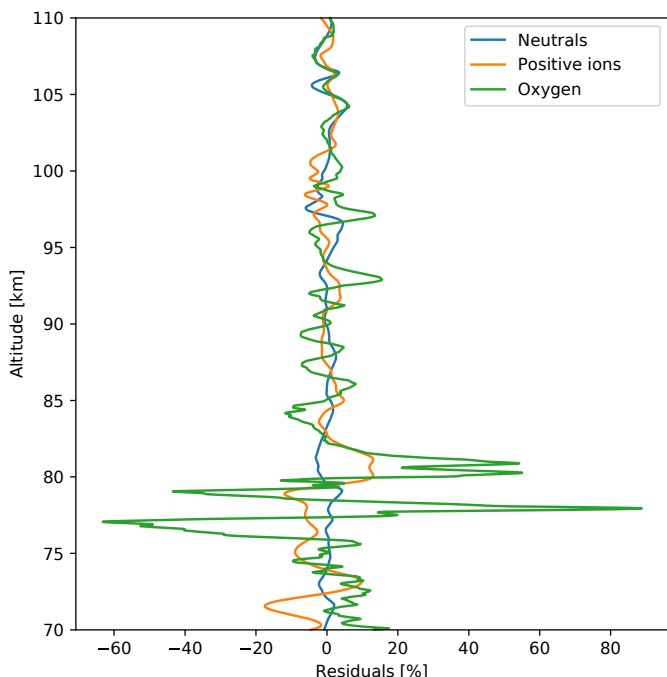

**Figure 11.** Relative density fluctuations (=residuals) of neutral air, positive ions, and atomic oxygen shown by blue, orange, and green profiles, respectively. These fluctuations were derived from the density profiles measured on the downleg of WADIS-2 rocket flight by subtracting a running average over 5 km long vertical window.

In Fig. 11 we show relative density fluctuations (residuals) for neutrals, positive ions, and atomic oxygen in blue, orange, and green, respectively. These fluctuations were derived from the density profiles measured on the downleg of WADIS-2 rocket flight. All three profiles show wave-like oscillations with (vertical) wavelengths in the range 1 to 5 km that can be attributed to gravity waves. Between 103 and 110 km altitude all three profiles oscillate in phase. Below ∼103 km height these density

fluctuations reveal similar wavelengths but shifted in phase relative to each other.

The density fluctuations of neutral gas shown in blue are used to derive turbulence energy dissipation rates, $\varepsilon$, as mentioned in Sec. 2. The resultant $\varepsilon$-profiles for up- and downleg are shown in Figs. 12a and 12b, respectively.

Blue and green profiles represent turbulence derivation utilizing different spectral models, i.e. of Heisenberg (1948) and Tatarskii (1971), respectively (see also e.g., Lübken et al., 1993). The difference between the green and the blue values can

be considered as method's uncertainty. The black bold and grey profiles show climatologies, that is mean seasonal values for winter (Lübken, 1997) and summer (Lübken et al., 2002), respectively. Fig. 12 reveals, for instance, that we observed turbulence activity in the entire altitude range from 60 up to 100 km. This is a characteristic feature for winter season. Also the upleg and downleg turbulence data qualitatively agree with each other except small region observed between 63 and 67 km on downleg. Vertical thick lines in Fig. 12 show mean values over the marked altitude regions (i.e., 10 km altitude bins). If

compared with results of our previous rocket campaign WADIS-1 that was conducted in summer (Strelnikov et al., 2017), the

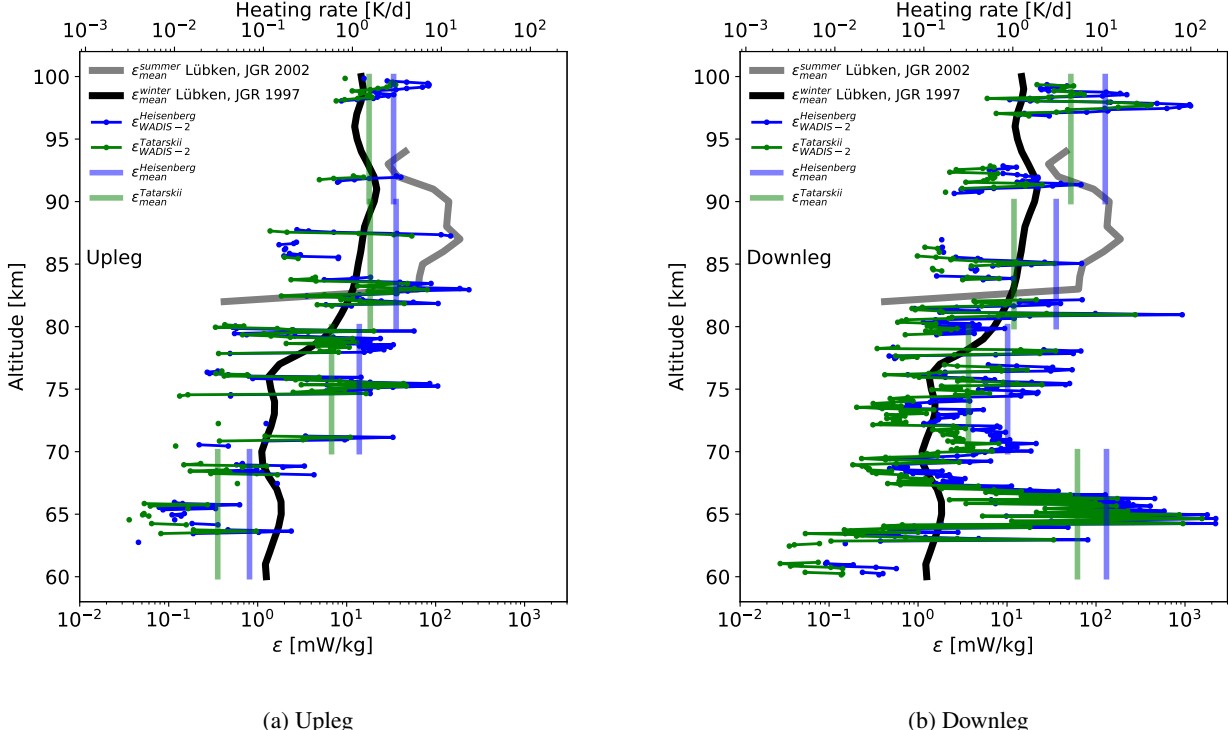

| (a) Upleg | (b) Downleg |

**Figure 12.** in-situ turbulence measurements by the WADIS-2 rocket launched on 1st of March 2015 at 01:44:00 UTC. Blue and green colors represent turbulence energy dissipation rates, $\varepsilon$, derived using spectral models of Heisenberg (1948) and Tatarskii (1971), respectively. Vertical lines show mean $\varepsilon$-values over the 10 km height bins. Bold black and gray profiles show turbulence climatologies for winter and summer, respectively.

observed winter turbulence field does not show big difference between up- and downleg measurements. If taking only mean values with their uncertainties in consideration, the height region 70 to 100 km looks very similar on both up- and downleg. The downleg turbulence measurements, however, reveal a patch of very strong turbulence around ∼65 km height. Turbulence filed in summer MLT observed during WADIS-1 campaign showed large oscillations in both space and time so that even

5  mean $\varepsilon$-values of up- and downleg rocket measurements differed by order of magnitude (Strelnikov et al., 2017). Turbulence variability in time was studied by analyzing MAARSY and EISCAT (European Incoherent SCATter Scientific Association) radar measurements which were properly scaled based on in-situ data. The MAARSY is only capable of measuring MLT parameters if some radar echo occurs, that is it needs presence of PMSE or PMWE. PMSE occurrence rate as observed by MAARSY is close to 100 % (Latteck and Bremer, 2017) which makes it easy to study MLT in summer season. The winter

10  echoes, PMWE, are much more rare, i.e. their occurrence rate is at most 30 % (Latteck and Strelnikova, 2015).

In Fig. 13 we show wavelet spectrogram of neutral density fluctuations shown in Fig. 11. This spectrum was used to derive the $\varepsilon$-profile shown in Fig. 12b (see Strelnikov et al., 2003, for details). We note that qualitatively similar picture in terms of

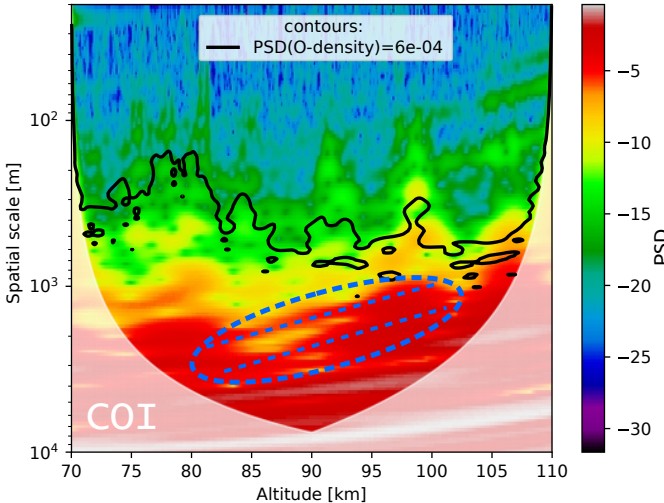

**Figure 13.** Wavelet spectrum of neutral density fluctuations as color contours. Black contour shows a constant-power line taken from wavelet spectrum of atomic oxygen density fluctuations (not shown here). White shading shows cone of influence (COI). Dashed oval and two lines mark region where power peak shifts from large to small-scales which can be attributed to a GW-saturation process.

turbulent structures could be inferred by analyzing the upleg data which is not shown here. In this spectrogram (Fig. 13) power falls down along the spatial scale axis from large to small scales. Noise level corresponds to bluish colors. Turbulent layers can be recognized in this figure as regions of green color which are extended to small-scales of the order of 100 m and less. Note that this only gives a rough approximate visualization of the turbulence structure. A more close and detailed examination of individual spectra (power spectral densities, PSD, vs frequency) is required to derive the $\varepsilon$-profile. Nevertheless, colored wavelet spectrograms like those shown in Fig. 13 help to identify power change at different scales in the spectrum. So, one can identify, for instance, two near parallel slopes in red color between $\sim$80 an $\sim$98 km that extend approximately from scales of 5 to 3 km down to 1 km and slightly below. This region is marked with the blue dashed oval and tilted dashed lines in Fig. 13. This picture is reminiscent of a GW-saturation process when vertical wavelength of GW becomes shorter (see, e.g., Fig. 5.3 in Nappo, 2002). Consistently, we observed a turbulence layer on top of this saturation region, i.e. in 98 to 100 km range.

In Fig. 13 we further compare the neutral density spectrum derived from the CONE measurements with the spectrum of atomic oxygen density fluctuations derived from the FIPEX measurements. The black contour on top of the colored scalogram is a constant power line taken from a similarly derived wavelet spectrum (not shown here) of the O-density fluctuations which are shown in Fig. 11. The PSD-value that represents the O-spectra (black line) lies well withing inertial subrange (i.e., $k^{-5/3}$ part) inside the regions where turbulence was observed. The contour line reproduces the small-scale structure of neutral density fluctuations quite well. So, all regions where neutral density structures extent to small-scales are also present in the O-density spectra. This suggests that atomic oxygen is affected by turbulent mixing.

In order to directly compare the spectral content of neutral air and atomic oxygen fluctuations we further show in Fig. 14 two dimensional slices of the corresponding wavelet spectrograms (i.e., for neutrals and oxygen). Fig. 14 shows spectra of neutrals

and O in black and blue colors, respectively. Additionally, in Fig. 14a we show a Fourier spectrum of the horizontal wind fluctuations (magenta color) shown in Fig. 10. The wind spectrum only extends to ∼200 m which is due to the limited altitude resolution of these measurements. The red dashed line marks slope of $k^{-3}$ (k is wavenumber) which is commonly attributed to gravity waves (Fritts and Alexander, 2003). It is clearly seen that the spectrum of wind fluctuations closely follows the $k^{-3}$ power law. Spectrum of neutrals is also very close to it down to ∼15 m scales where white nose produced by instrument electronics dominates the signal. The spectrum of the instrumental noise is marked in Fig. 14 by a bold gray horizontal line. Dashed green line in Fig. 14a marks $k^{-5/3}$ slope, that is the famous Kolmogorov power law which suggests that turbulence was acting on tracer distribution. It can be seen that the spectrum of O-density fluctuations might have been affected by turbulence.

Fig. 14b shows spectra of neutral and oxygen density fluctuations taken from the region where moderate turbulence was observed near 82 km. Dark blue solid bold line also shows the fitted theoretical model of Heisenberg (1948). This model only describes two parts (or subranges) of turbulence spectrum. Its left part called inertial subrange follows the $k^{-5/3}$ power law whereas the dissipation part often referred to as viscous subrange is described by $k^{-7}$ slope. One can see that the leftmost part of both spectra (i.e., for O and neutrals) follows the $k^{-3}$, i.e. GW-like slope (also marked by red dashed line) which then smoothly transits into the turbulence spectrum well described by the Heisenberg model. The inertial subrange of both spectra spans over the same range of spatial scales, namely from the so-called outer scale $L_B$ ≈500 down to the inner scale $l_o$ ≈30 m. We recall here, that the $\varepsilon$-value is directly derived from the spatial scale $l_o$ defined as transition between inertial and viscous subranges.

Fig. 14c shows spectra from region just below that one described above. The $\varepsilon$-value here drops by one order of magnitude and the inertial subrange shrinks to ∼100 m. Interestingly, the atomic oxygen spectrum reveals a somewhat different shape. Fig. 14b and 14c demonstrate two types of spectral behavior of O-density fluctuations inside turbulence layers observed during WADIS-2 rocket experiment.

## 5 Discussion

In this section we discuss the shown above fluctuations in different quantities measured with rocket-borne and ground-based instruments. First we focus on the large scale morphology in temperature field. Then we discuss fluctuations which might be attributed to gravity waves. Finally, we consider the observed smallest-scale structures due to turbulence.

Fig. 7 shows a prominent temperature enhancement around ∼100 km altitude. As mentioned in Sec. 3, Szewczyk et al. (2013) observed similar temperature enhancement in the same altitude region. They showed, that this signature was observed by MLS satellite instrument over large region and was also seen in lidar data for long time. These two facts lead them to conclude that the observed temperature enhancement could be qualified as mesospheric inversion layer (MIL). Their lidar temperature measurements also showed that MIL descended together with tide. A striking feature accompanied MIL studied by Szewczyk et al. (2013) was a very strong turbulence layer with $\varepsilon$=2 W/kg (equivalent to a heating rate of 200 K/day) observed on top of their upper MIL. Interestingly, our downleg data reveal very similar situation. Namely, on top of the temperature enhancement we also observe a vigorous turbulence layer with $\varepsilon \simeq 1$ W/kg (or ∼100 K/d). Our temperature enhancement (compare Fig. 7

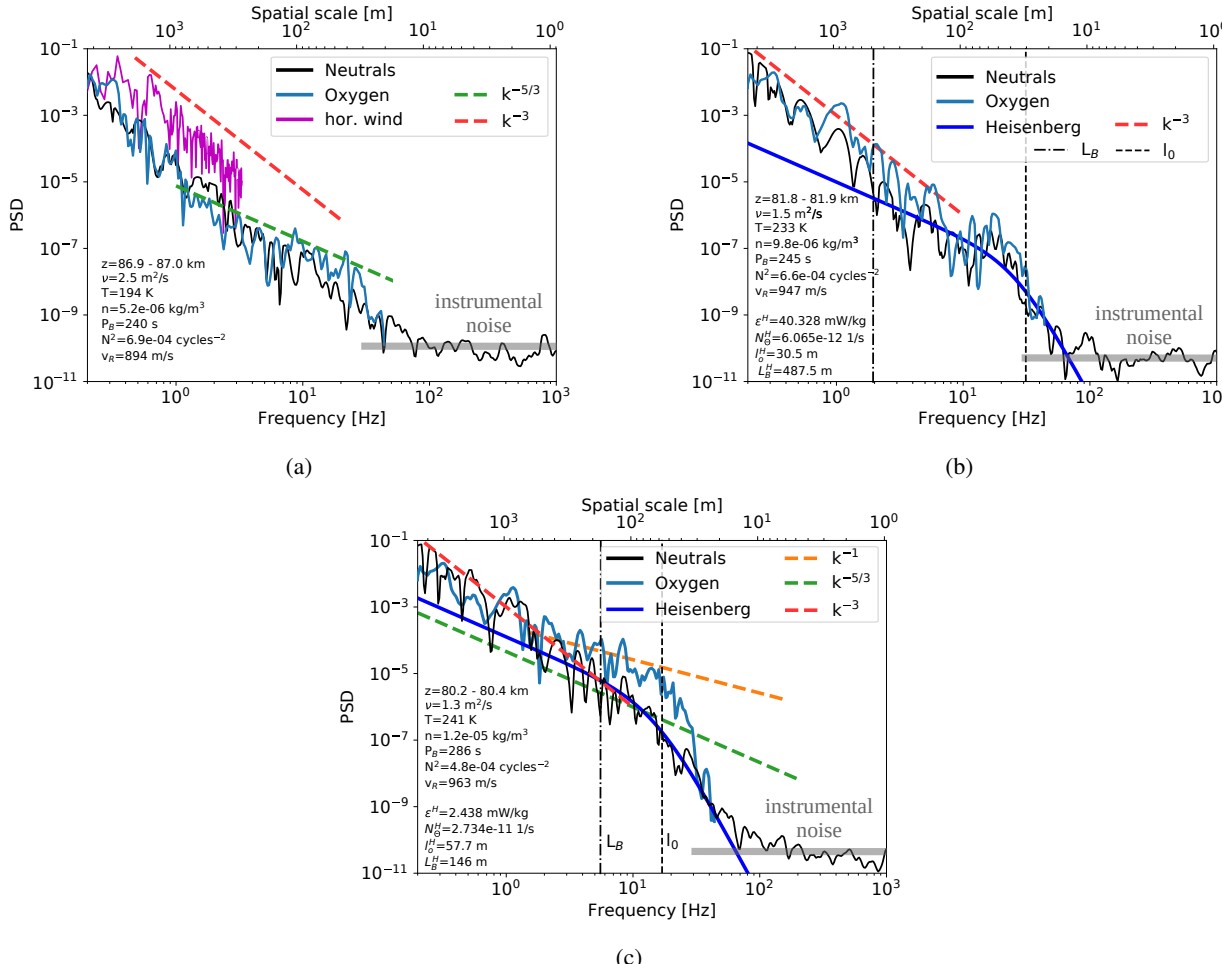

**Figure 14.** Normalized power spectral densities (PSD) of horizontal wind fluctuations (magenta), $\Delta N_n/N_n$ (black), and $\Delta[O]/[O]$ (blue) measured during descend of WADIS-2 sounding rocket. Dashed red, green, and orange lines show slopes with $k^{-3}$, $k^{-5/3}$, and $k^{-1}$ power law, respectively. Vertical black dashed and dashed-dotted lines mark inner and buoyancy scales of turbulence, $l_\circ$ and $L_B$, respectively. Bold gray horizontal lines mark instrumental noise. a) All spectra reveal $k^{-3}$ slope attributed to gravity waves; b) All spectra reveal both waves and turbulence. The neutral- and O-spectrum are near identical. c) Small-scale part of the O-spectrum reveal a $k^{-1}$ slope.

and 3) also descends within a time period of several hours. Their temporal behavior is studied in Wörl et al. (2019), who found dominant periods of 24, 12, and 8 hours. Its spatial extent is at least 60 km (distance between ascent and descent of the sounding rocket plus the distance to the south-east Na-lidar beam, see Fig. 1). The upleg rocket data, however, are somewhat different. While the upleg measurements show similar temperature enhancement, the turbulence observed is one order of

magnitude weaker. Szewczyk et al. (2013) argued that if turbulence itself did not produce the MIL, it might have amplified it. This statement is consistent with our observations. The local temperature maximum near 100 km reveals ∼10 K lower values for upleg measurements than for downleg (see Fig. 7). Also turbulent energy dissipation rates measured on upleg show ∼one oder of magnitude lower values than those observed during rocket descend (Fig. 12).

      In the previous sections we showed ground-based wind and in-situ density and temperature measurements which reveal

signatures of gravity waves. Closer comparison of density fluctuations of neutral gas, positive ions, and atomic oxygen in Fig. 11 shows at least two things. First, the amplitudes of GW in these tracers are different. Second, the phases can be shifted relative to each other. Fritts and Thrane (1990) and Thrane et al. (1994) studied the relationship between ion and neutral density fluctuations due to gravity wave and turbulent motions both theoretically and experimentally. Their results indicated that this relationship in the middle atmosphere ranges between an adiabatic and a chemical equilibrium limit, depending on the

characteristics of the wave or turbulent motion and the ion production and recombination rates. In pure adiabatic limit case, that is when fully developed turbulence dominates in observed timeseries, the neutral and ion density fluctuations must be in anti-phase, i.e. shifted by $\pi$ relative to each other. Fritts and Thrane (1990) also noted that we should not expect the fluctuations of ions and neutrals to be either in or out of phase. Rather, they will have a phase shift that depends on the relative values of the intrinsic wave frequency, the mean ion number density, and the effective recombination rate.

Fritts and Thrane (1990) showed that the phase shift between fluctuations of different species depends on the relative magnitudes of wave and chemistry time scales. For low frequency waves this phase shift $\rightarrow 0$, indicating that the species are in chemical equilibrium. This situation is well observed just above the MIL-like temperature enhancement in the altitude range from 102 to 110 km (see Fig. 11), where all three species, i.e. oxygen, ions, and neutrals show nearly the same oscillations. This behavior is observed directly above the upper turbulence layer measured by rocket-borne instruments on both up- and

downleg.

      Next, we focus on the feature seen in the wavelet scalogram (Fig. 13) which is marked by the dashed lines. It suggests that waves with vertical wavelength of ∼2–3 km saturated within altitude range ∼80 to 98 km and broke producing turbulence layers (Fig. 12). That is, our new measurements suggest that turbulence produced by breaking GW amplifies temperature fluctuations producing MIL-like signature in temperature field. Note that density fluctuations in Fig. 11 do not have this large

MIL-like signature because of the background subtraction procedure which filtered out waves longer than 5 km.

      It is interesting to note that also the altitude region below ∼82 km reveals feature, similar to what is observed near 100 km height: stronger turbulence activity on downleg accompanies higher temperatures. Furthermore, right above this active region GW-amplitudes seen in temperature data (Fig. 7) are very low. $\varepsilon$-profiles (Fig. 12) also demonstrate lower turbulence activity (which might be connected to low GW-amplitudes). Also wind fluctuations (Fig. 10) are consistent with this picture. Their

amplitude grows in the height region 30 to 70 km and afterwards, near 85 km it vanishes.

All these facts together suggest that we observed waves saturation and dissipation in the altitude range 70 to 80 km accompanied by turbulence production. Above these heights we observed GW-amplitude growth and breakdown near 100 km. The latter can be either attributed to secondary GW generated by turbulence (see e.g., Becker and Vadas, 2018, and references therein) or just another wave package passing through our observation field.

The spatial scales considered in the discussion above are of the order of 1 to few kilometers. This range of vertical scales is characteristic for GW in MLT. If we look at these fluctuations in spectral domain (e.g., shown in Fig 14) we see that they follow the $k^{-3}$ slope, usually attributed to GW (see e.g., Smith et al., 1987; Weinstock, 1990; Fritts and Alexander, 2003; Žagar et al., 2017, and references therein), in the range of scales from 5 km down to hundreds of meters. Now we discuss structures at smaller scales. A power increase in spectrum which appears at spatial scales below $\sim$500 m must rather be attributed to action

of turbulence. If the spectrum reveals a clear shape which can be mathematically described by a model (e.g., Heisenberg, 1948; Tatarskii, 1971) we argue that we observed an active turbulence and derive its energy dissipation rate, $\varepsilon$. To do so, we have to make sure, that the density fluctuations that we use for turbulence analysis are passive, that is do not influence the flow and conservative, i.e. its value is not affected by the flow (see e.g., Lübken, 1992, 1997; Lübken et al., 1993, 2002). It was also shown, that plasma density fluctuations can both satisfy this requirement and, under specific conditions, can be affected by

non-turbulent processes like enhanced recombination of electrons with cluster ions (e.g., Röttger and La Hoz, 1990), the effect of charged ice particles in PMSE (e.g., Cho et al., 1992), or plasma instabilities (e.g., Blix et al., 1994; Strelnikov et al., 2009). If spectra at small-scales reveal some enhanced power which cannot be described by a model, it might imply that we observed a residual structuring after action of turbulence some time before. In such a case we cannot derive any parameter of turbulence from this data.

Recalling to above discussion about relationship between density fluctuations of neutrals and ions we note that our analysis results suggest that a similar relationship should also exist between small-scale atomic oxygen density fluctuations and other constituents, i.e. ionospheric plasma and neutral gas. So, e.g., earlier common volume rocket-borne measurements of atomic oxygen and electron densities ($N_e$) by Friedrich et al. (1999) showed a clear correlation between fine structure of O- and $N_e$-density profiles.

In the height range from $\sim$78 to $\sim$85 km in Fig. 11 fluctuations of ion density are in phase with those of O-density and they both are almost in anti-phase with the neutrals. This might imply that we here observed O-densities which are in chemical equilibrium with ion densities (phase shift close to zero) located inside turbulence layers (phase shift $\sim \pi$). The turbulence measurements indeed show turbulent layers in this altitude range (see Fig 12b). On the one hand, this supports the theory of Fritts and Thrane (1990), but at the same time it raises the question how O-density behaves inside turbulence and whether

turbulence affects the height distribution of atomic oxygen.

    Now we discuss the power spectra shown in Fig. 14. This figure demonstrates three types of spectra found in the fluctuations measured during the WADIS-2 flight. As noted above, the panel (a) shows spectra typical for gravity waves whereas two other panels, (b) and (c), demonstrate turbulent spectra. The both spectra in panel (c), i.e. the PSD($\Delta N_n/N_n$) and the PSD($\Delta[O]/[O]$) shown with black and blue lines, respectively, are near identical, suggesting that at these altitudes the chemical time constant

is larger than the turbulent one. At the same time, just below this height, that is where the chemical time constant is expected to be comparable, the PSD($\Delta$[O]/[O]) shows different spectral behavior revealing a k$^{-1}$ slope at small-scales (Fig. 14c).

Detailed analysis of all the regions where different types of O-density spectra occur is rather difficult because of different reasons. One reason is that when turbulence is very strong, its spectrum extends down to scales below current FIPEX resolution limit of $\sim$20 m, that is the O-density spectrum appears not fully resolved. So far we can only identify three regions where third type of spectra (like in Fig. 14c) was observed in the downleg measurements: between 90.9–91.2 km, 80.2–80.4 km, and 81.1–81.7 km.

Let us now discuss which processes could potentially contribute to the change of diffusive properties of atomic oxygen as shown in Fig. 14c. Since satellite observations show that horizontal (i.e., meridional and zonal) gradients of atomic oxygen at 70–100 km are much smaller than the vertical one (e.g., Mlynczak et al., 2013, 2018), it is reasonable to assume that meridional and zonal transport of atomic oxygen can be neglected in this case. Characteristic time due to vertical advection is of the order one month (e.g., Brasseur and Solomon, 2005), hence, it is also negligible compared to e.g., turbulent mixing. In order to clarify relative significance of the chemical processes we compare characteristic times derived from in-situ measurements in Fig. 15. The chemical lifetime of atomic oxygen, $\tau_O^{chem}$, shown in Fig. 15 by the blue line, was derived as reciprocal of loss

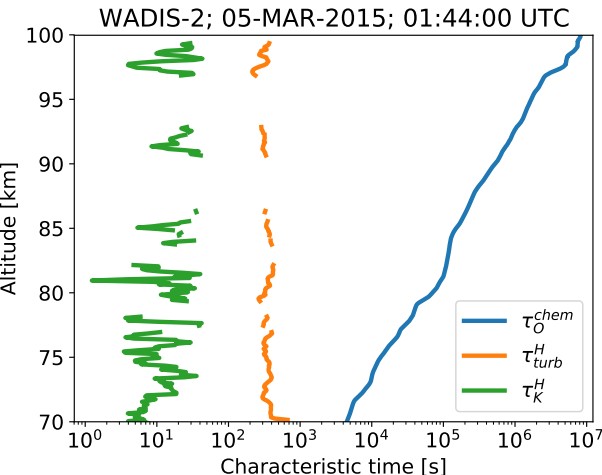

**Figure 15.** Characteristic time scales derived from in-situ measurements. $\tau_O^{chem}$ - chemical lifetime of atomic oxygen, $\tau_{turb}^H$ - characteristic time scale for largest turbulent eddies derived based on Heisenberg model, $\tau_K^H$ - Kolmogorov time scale derived based on Heisenberg model.

term (Brasseur and Solomon, 2005, Eq. 5.30), omitting the loss due to reaction with ozone because in mesopause region it is not significant (Smith et al., 2008). The Kolmogorov time scale, $\tau_K^H = (\nu/\varepsilon)^{1/2}$, where $\nu$ is kinematic viscosity of air, was derived using results obtained with the model of Heisenberg (1948), i.e. $\varepsilon_{WADIS-2}^{Heisenberg}$ shown in blue in Fig. 12. The characteristic time scale for largest turbulent eddies, $\tau_{turb}^H = L_B/\omega_{turb}$, where the turbulence buoyancy scale which characterizes the largest

turbulence eddies was derived based on Weinstock (1978) as:

$$L_B = \frac{2\pi}{0.62} \sqrt{\sqrt{\frac{\varepsilon}{N^3}}}$$

where $N$ is the buoyancy frequency and the mean turbulent velocity $\omega_{turb} = (\varepsilon/(N*0.49))^{1/2}$ that characterizes advection by the largest eddies was calculated as derived by Weinstock (1981) and using results obtained with the model of Heisenberg (1948), i.e. $\varepsilon_{WADIS-2}^{Heisenberg}$. The Kolmogorov time scale characterizes motion of the smallest turbulent eddies (i.e., of the size of Kolmogorov scale, $\eta = (\nu^3/\varepsilon)^{(1/4)}$) and, therefore the range ($\tau_K^H$, $\tau_{turb}^H$) describes all turbulent motions. Fig. 15 thereby shows, that during the WADIS-2 rocket flight turbulent advection was much faster than chemical processes relevant for atomic oxygen. Such situation, however, is most probably typical for night-time MLT at high northern latitudes.

Friedrich (2016) noted, that O-density should positively correlate with $N_e$ for two reasons: (1) [O] inhibits the formation of water cluster ions (e.g., $H^+(H_2O)_n$) which have significantly faster recombination rates than molecular ions ($O_2^+$, $NO^+$) (2) [O] provides a reverse reaction effectively detaching electrons from negative ions. In terms of density fluctuations, $N_e$ is ultimately connected to the ambient ion density, $N_i$, which, in turn at heights below ∼100 km are essentially governed by collisions with neutrals (e.g., Rapp et al., 2003). That is the small-scale structures produced by turbulence must be observed in these species too. The difference in their response to turbulence arise due to difference of their diffusivity. Thus, diffusion constant of charged constituents is affected by ambipolar forces resulting from electrostatic coupling between positively and negatively charged species (e.g., Chen, 2016). Moreover, it was shown that heavy charged aerosols if present can significantly reduce plasma diffusivity allowing extension of eddy cascade in those species down to much smaller scales (e.g., Rapp et al., 2003). This, for instance can lead to such phenomena like polar mesosphere radar echoes in summer or winter, PMSE or PMWE, respectively (see e.g., Rapp and Lübken, 2004; Lübken et al., 2006). Under conditions when diffusivity, D, of plasma species is considerably reduced its spectral behavior can be described by a model which includes this parameter via e.g., Schmidt number, Sc=$\nu$/D, where $\nu$ is kinematic viscosity of ambient gas (e.g., Driscoll and Kennedy, 1985). Such models implement theory of Batchelor (1958) and describe the small-scale part of the spectrum by a $k^{-1}$ power law. In other words, if spectrum of density fluctuations reveals the $k^{-1}$ slope at scales smaller than those where $k^{-5/3}$ is present, this means that the diffusivity of this constituent is reduced. To summarize, spectra of O-density fluctuations which show the $k^{-1}$ slope (one example of those is shown in Fig. 14c), may imply that atomic oxygen can show different diffusion properties inside turbulence layers. That is, addressing the question whether turbulence affects height distribution of atomic oxygen, we can definitely say yes. The observed $k^{-1}$ spectral behavior, however needs further in depth investigation and lies out of scope of this paper.

## 6  Conclusions

In this paper we present an overview of the entire scope of measurements conducted in the frame of the WADIS-2 sounding rocket campaign. We also demonstrate the important role of small-scale processes like gravity waves and turbulence in the distribution of O in the MLT region.

We show that all measured quantities, including winds, densities, and temperatures, reveal signatures of both waves and turbulence. Analysis of density fluctuations measured by rocket-borne instruments supported the theory by Fritts and Thrane (1990), but also suggested that a similar relationship might exist between atomic oxygen and other constituents, i.e., plasma and neutrals.

5    Atomic oxygen inside turbulent layers showed two different spectral behaviors at scales smaller than $\sim$300 m. Some of the O-density spectra reproduce spectra of neutral gas, but some of them show a $k^{-1}$ slope. A more detailed study of such very small scales in O-density data is subject of our future work. In particular, a somewhat higher altitude resolution and enhanced sensitivity of FIPEX sensors may yield more detailed picture for our future rocket experiments.

*Author contributions.* M.R., F.J.L. and B.S. designed and directed the project; B.S., M.F., R.L., J.H., M.K., J.G., S.L., S.F., G.B., J.H., A.B.,

10    M.J.T. designed and directed the subprojects and related instruments; M.E., M.F., J.H., M.K., G.B., B.P.W., H.A., R.L., J.H., R.W., A.B., M.J.T, P.D.P performed the experiments; B.S., M.E., M.F., J.H., G.B., B.P.W., T.S., H.A., I.S., R.L., M.G., J.H., R.W., A.B., M.J.T, P.D.P analyzed the data; All authors contributed to the final manuscript.

*Competing interests.* The authors declare that they have no conflict of interest.

*Acknowledgements.* This work was supported by the German Space Agency (DLR) under grant 50 OE 1001 (Project WADIS). Authors

15    thank DLR-MORABA for their excellent contribution to the project by developing the complicated WADIS payloads and campaigns support together with the Andøya Space Center. Authors thank H.-J. Heckl and T. Köpnick for building the IAP rocket instrumentation. The design and initial development of the AMTM was supported under an AFOSR DURIP grant to USU. The AMTM installation and operations at ALOMAR were supported under the NSF collaborative grant AGS-1042227. The sodium lidar observations and data analysis were supported by NSF grants AGS-1136269, AGS-1734693, and AGS-1829138.

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
