# Peer review of "Simultaneous in-situ measurements of small-scale structures in neutral, plasma, and atomic oxygen densities during WADIS sounding rocket project"

_Atmospheric Chemistry and Physics, 2018_

## Referee Comment (RC1) · Anonymous Referee #1 · 21 Feb 2019

articleamsmath amssymb graphicx

**Review Report**

**1   General Comments**

Strelnikov et al. present the results from measurements conducted during the WADIS-2 rocket campaign on 05 March 2015 at 01:44:00 UTC to investigate the small-scale

structures like gravity waves and turbulence in neutral, plasma, and atomic oxygen densities. Fluctuations of these densities are used to understand the small-scale dynamical behaviour of MLT region. This study is interesting since rocket estimated mean turbulence kinetic energy (TKE) dissipation rate seems to be following the similar trend of mean TKE dissipation rate of winter season in most of the height regions when there was no radar echoes in MAARSY. The results are relevant and it will contribute to the understanding of turbulence activity and energetics in MLT. It discusses the physical and chemical processes governing the energetics of the MLT region. It can be accepted for publication after the revision. I suggest for major revision. Strelnikov et al. claims that atomic oxygen fluctuations are generated by small-scale dynamics such as gravity waves and turbulence; they also try to convince that atomic oxygen fluctuations are not associated with charge fluctuation through chemical reactions with help of no-echoes case in MAARSY radar. If observed fluctuations are purely governed by turbulence then those can be compared with the low-latitude turbulence intensities for enhancing understanding on mesospheric turbulence. Qualitative results are presented but results are not enough with quantitative and clarity statements. They mention rough words like some, somewhat in many places those should be replaced with quantitative results. Notably, references, methodology and assumptions are not mentioned in many places which are figured out in comments.

**2   Specific Comments**

1. Page 1, lines 12-13: 'GW might dissipate and thereby generate turbulence'-GW might break and generates turbulence. This statement can be supported by appropriate references of observational (e.g., https://doi.org/10.1002/2015JD024283) work(s) since a lot of studies are available but nothing cited.

2. Page 1, lines 13-14: 'Turbulence mixing and redistribution of trace constituents' can be supported by appropriate references of observational work(s).

3. Page 2, lines 6-7: 'chemical heating rates in the mesopause region of several K/day which is comparable (or even competitive) to those of turbulent heating' can be supported by appropriate references of observational work(s) whos study concluded it statistically and/or mention case study.

4. Page 4, lines 23-25: It is stated in Page 3, lines 13-14 that 'MAARSY operated continuously to detect PMWE echoes if any' but no echo is detected during the night of rocket launch. It can be mentioned that echo occurrence depends on both nature of the target and sensitivity of the radar. It is better to reveal with help of volume reflectivity map (without any signal threshold) during that night of rocket launch. And select the profile of volume reflectivity to discuss the sensitivity of MAARSY at that time of sounding in MLT since this article deals with small scale dynamical fluctuations in densities while avoiding chemical reactions. It can be mentioned that during February-March 2011/2012 and 2012/2013, PMWE of MAARSY appeared throughout day and night with dis-continuities in seasonal-local time variation (see Fig. 3 of Latteck and Strelnikova, 2015).

5. Page 4, line 27: 'temperature field measured by the lidars'. Is it measured or estimated? If temperature is estimated then provide appropriate references to methodology for temperature retrieval which is used here.

6. Page 5: In Fig. 2, 'temperature of RMR- and Fe- lidar are combined'. How do they agree?

7. Page 5: In Fig. 2 and Fig. 3, It is better to limit the steps in colormap for easy reference of temperature in Fig. 2 and Fig. 3. And also provide temperature labels in colormap of Fig. 3. And show the results Fig.2 and Fig. without interpolation.

8. Page 7: In Fig. 5, Rocket observed densities are seem to be very much smoothed. How it is smoothed?

9. Page 7: In Fig. 5, Do you find any spin frequency in up- and down-leg of rocket observations? If yes, what are the methods are used to remove those frequency and what are they?

10. Page 7: Line 14, 'Both up- and down-leg profiles are very similar interms of oscillations'. It is worth to show the detrended densities or density-normalised density fluctuations in subplot along with Fig. 5. item Page 7: Line 17, 'some GW-signatures'. It is better to quantify the GW amplitudes corresponding to Fig. 6, as you like, below 80 km, 83-90 km and 95-100 km.

11. Page 8: In Fig. 6, It is worth to describe the mothod(s) and along with valid assumptions used for temperature retrieval from densities. Include the profiles of temperature from Lidars in Fig. 6 for ready comparison of temperature oscillations between remote sensing and in-situ measurements.

12. Page 9: Line 1, 'some aurora was seen'. It is better to mention the wave length of those observed emissions.

13. Page 9: Line 14, 'O-density profiles reveal some oscillations'. It is better to show profiles of density-normalised density fluctuations in subpot along with Fig. 8.

14. Page 11: Fig. 10, It is better to have percentage of amplitude and density-normalised density fluctuations in Fig. 10.

15. Page 11: Line 9, Before comparison of results, describe the observed results of turbulence intensity from different techniques (Heisenberg and Tatarskii) to demostrate the need of different techniques here. And also present the mean of them in order to present quatification of turbulence intensity.

16. Page 12: Lines 6-7, 'This picture is reminiscent of a GW-saturation process when vertical wavelength of GW becomes shorter'. It is worth to quantify the GW activity using the lidar observations and compare them with Fig. 12 for quantitative and qualitative statements since this article mainly focus on dynamics of small-scales.

17. Page 13: Line 14, Briefly describe the assumptions behind this Heisenberg model and the value of its constants which is used in it. How those constants are obtained?

18. Page 14: Fig. 13, Fig. 13a,b & c show the spectra of neutral densities in black line. In frequencies more than 100 Hz, neutral density fluctuations are appearing as almost flattened. Is this flattening due to instrumental noise? Indicate the instrumental noise as a line in Fig. 13a,b & c?

19. Page 15: Lines 1-2, '$\epsilon$-value is directly derived from the spatial scale $l_o$'. Provide the formulea and appropriate references along with valid assumptions and constants.

20. Page 15: Line 15, 'MIL descended together with tide'. Quantify the tidal activity using the lidar temperature measurements since demostrated examples behave very good.

21. Page 15: Lines 18-19, 'This temperature enhancement also descends within a time period of several hours'. Provide the descend rate.

22. Page 15: Line 20, 'The upleg rocket data, however, are somewhat different'. What are the differences are observed and at what altitude?

23. Page 15: Line 22, 'MIL, it might have amplified it'. what are the sources might cause amplification of MIL at this altitude (provide appropriate references also)?

24. Page 15: Line 23, 'The weaker turbulence on upleg accompany âĹij10 K colder temperature maximum'. I could not understand this. Can you rewrite it to make understanding in simple way.

25. Page 15: Line 26-27, 'First, the amplitudes of GW in these tracers are different. Second, the phases can be shifted relative to each other'. Can you provide values and discuss it? Since you have unique observations.

26. Page 15: Lines 31-32, 'the neutral and ion density fluctuations must be in anti-phase'. Is it possible to quantify the angle between neutral and ion fluctuations ($< a, b > = |a|.|b|.cos(angle)$) for those three altitudes or altitude regions where coherence exist? Then, Provide the quantification of those results and discuss them.

27. Page 16: Lines 6-7, 'where all three species, i.e. oxygen, ions, and neutrals show nearly the same oscillations'. It can be discussed based on quantified angles.

28. Page 17: Line 35, 'the $k$-1 slope at scales smaller than those where $k$-5/3 is present'. Provide the profiles of outer scale and inner scale as like $\varepsilon$ and indicate the altitudes in it where the spectral slope is seen as $k$-1. It provides the active altitude regions where two different type of diffusions take place since it is a unique measurements to deal with. And also compare these turbulence parameters with the same of low-latitude turbulence parameters since day-time low-latitude mesospheric turbulence measurements not have any affect with dusty plasma.

**3 Technical Corrections**

1. Care can be taken thoughout article to write 'in-situ' instead of 'in situ'. And also introduce 'intend in every first line' of every section.

[Figure]

2. Page 1, line 1: 'In this paper, we' instead of 'In this paper we'

3. Page 2, line 10: 'mainly' instead of 'manly'

4. Page 2, line 14: 'is the region' instead of 'is region'

5. Page 2, line 14: 'persistent turbulence field plays crucial role in global circulation' can be explained briefly.

6. Page 3, line 10: I couldn't find the expansion of 'RMR-'.

7. Page 3, line 15: 'in case if they occur' instead of 'in case they should occur'

8. Page 7: Line 12, 'temperatures, respectively' instead of 'temperatures.'

9. Page 7: Line 19, '100 km and, it is' instead of '100 km and is'

10. Page 8: Line 19, 'In Fig. 7, we' instead of 'In Fig. 7 we'

11. Page 9: Line 18, 'In the next section, we' instead of 'In the next section we'

12. Page 10: Line 16, 'In Fig. 10, we' instead of 'In Fig. 10 we'

13. Page 11: Line 14, 'In Fig. 12, we' instead of 'In Fig. 12 we'

14. Page 11: Line 14, 'fluctuations which is shown' instead of 'fluctuations shown'

15. Page 12: Line 1, 'Note that' instead of 'Note, that'

16. Page 12: Line 9, 'In Fig. 12, we' instead of 'In Fig. 12 we'

17. Page 13: Fig. 12, It is better to project the altitude in y-axis and scales in x-axis to compare easily with other results.

18. Page 13: Fig. 12, Briefly describe, how is the wavelet spectrum obtained? what type of wavelet is used and frequency of it?

19. Page 13: Line 3, 'fluctuations, we' instead of 'fluctuations we'

20. Page 13: Line 5, delete 'in Fig. 13a'

21. Page 13: Line 6, delete 'shown'. And write 'Fig. 13a which is corresponding to the fluctuations presented in Fig. 9' instead of 'Fig. 9'

22. Page 14: Fig. 13, Fig. 13 has been categorised as a, b & c based on nature of spectra in caption of Fig. 13, as assumed to be common volume/altitudes in question. Actually, different nature of spectra are seen at different altitude, so better to high-light the altitude in figure caption and title of the figure as (a) —- km, (b) —- km, and (c) —- km. And also indicate this height regions in Fig. 11a & b.

23. Page 15: Line 8, 'In this section, we discuss the results of those fluctuations' instead of 'In this section we discuss the shown above fluctuations'

24. Page 15: Line 12, 'They shown that' instead of 'They showed, that'

25. Page 15: Line 15, 'with the phase descend of tide' instead of 'with tide'.

26. Page 15: Line 24, 'sections, we shown' instead of 'sections we showed'

27. Page 16: Line 12, 'Note that' instead of 'Note, that'

28. Page 16: Line 14, 'note that' instead of 'note, that'

29. Page 16: Line 24, 'Fig. 13), we' instead of 'Fig 13) we'

30. Page 16: Line 28, '1971), we' instead of '1971) we'

31. Page 16: Line 28, 'its kinetic energy' instead of 'its energy'

32. Page 17: Line 14, 'Now we' instead of 'Now, we'

33. Page 18: Line 6, 'In this paper we' instead of 'In this paper, we'

---

## Referee Comment (RC2) · Anonymous Referee #2 · 25 Mar 2019

**1  General Comments**

This paper describes measurements taken during the WADIS-2 sounding rocket campaign. In particular, the effects of gravity waves and turbulence on densities of atomic oxygen, other neutrals, and ion species were studied. The authors found signs of waves and turbulence in all of their observations and were able to link regions of heating to gravity wave breaking and nearby layers of turbulence. This is a novel study describing an interesting new dataset that should add to the understanding of the physics

and chemistry of the MLT region. The paper is fairly well written, although there are a number of language faults (some of which are recorded below), and it was often required to hunt down explanations and methods in other sections or figure captions. I recommend publication in ACP once the comments below are addressed.

**2 Specific comments**

1. Page 2 line 8: provide citation for turbulent and solar heating rates.

2. Page 4 line 7: FIPEX - define acronym

3. Page 4 line 9: Could you give a brief description of this technique?

4. Figure 2: How are the RMR and Fe lidar data combined? What is the direction of these measurements?

5. Section 3.2, second paragraph: mention in the text (not just in the figure caption) which instrument was used to derive the total number density and temperature profiles and how the temperatures are derived from the densities.

6. Page 8 paragraph 2: What are the values of the ionospheric parameters used in the IMAZ model to produce the profile in Fig. 7?

7. "This is in accord with the fact that some aurora was seen" - by what instruments? "auroral emission detector... registered some auroral emission above 100 km" - how does this instrument work, what is measured? It might be useful to add this instrument to your list of rocket instruments in Sect. 2. Could you quantify "some auroral emission"?

8. Page 9, line 3: "The shown plasma density profiles yield relative density measurements and therefore they were normalised..." This sentence seems to be

the wrong way around: do you mean the measurements yield relative quantities that are therefore normalised to produce the profiles in Fig. 7? If so, this could also be mentioned in the caption.

9. Figures 5, 7, and 8: Please use consistent units: $m^{-3}$ is used in Figs. 5 and 7, and $cm^{-3}$ is used in Fig. 8.

10. Figures 7 and 8: In the captions, I am not sure I see what is the "same as Fig. 5", except that these are atmospheric density profiles, but they are from different species, measured by different instruments!

11. Figure 7: What is the quantity plotted from the IMAZ model? Is it ion or electron density?

12. Page 9 paragraph 2: I think you need to provide more details on the FIPEX instrument (either here or in Sect. 2). Could you specify why you are "mostly confident" in the descending data but not the ascending? Is this through comparison with data from the MISU photometers? Do you have an explanation as to what might be causing FIPEX to give erroneous measurements during one phase of the rocket flight but not another?

13. Page 9 line 12: Are you linking the increase in oxygen density seen by SABER above 100 km to auroral activity because of the auroral emission measured at these altitudes by the rocket-bourne MISU instrument? If so, why is a similar increase not seen in the FIPEX oxygen measurements taken at the same location as those from the MISU instrument?

14. Page 9 paragraph 3: Again, a little more information on what the "other FIPEX sensors" measure would be helpful!

15. Page 10 line 6: "small-scale stuff": colloquialism, please rephrase

16. Page 10 paragraph 3: please specify which data are used to determine the fluctuations in Fig. 10; are these up or down leg measurements?

17. Fig. 11: provide the references for the turbulence climatologies in the text (not just the figure legend)

18. Page 11 line 7: "Also the upleg and downleg turbulence data qualitatively agree with each other." — they do not agree below 70 km.

19. Page 11 line 9: "If compared with results of our previous rocket campaign WADIS-1 that was conducted in summer (Strelnikov et al., 2017), the observed winter turbulence field does not show big difference between up- and downleg measurements." It would be helpful to expand a little here and give (maybe just a sentence) on the WADIS-1 turbulence field.

20. Page 12 line 2: define acronym PSD, power spectral density

21. Page 12 line 6: "This picture is reminiscent of a GW-saturation process when vertical wavelength of GW becomes shorter." – could you provide a citation here?

22. Page 12 paragraph 1: how does the wavelet spectrum shown in Fig. 12 for the down leg compare to that of the up leg?

23. Page 12 line 9: are the O densities used here from the FIPEX instrument?

24. Figure 13: would it be possible to highlight the altitudes of each 2D slice forming the three panels? These could also perhaps be shown in Fig. 12.

25. Page 16 line 25: "usually attributed to GW" – could add a citation here

**3 English language corrections (not exhaustive)**

26. Page 1 line 9: ". . . MLT is host **to**. . ."

27. Page 1 line 13: "A part" (2 words)

28. Page 1 line 14: "**the** MLT"

29. Page 2 line 13: "that **the** mesopause region"

30. Page 2 line 14: "it is **the** region where. . . plays **a** crucial role"

31. Page 2 line 16: "it can affect" (remove 's')

32. Page 2 line 18: "concentration of atomic oxygen alongside the state of. . ." (remove 'with')

33. Page 2 line 28: "critically discuss our findings" (remove 'the')

34. Page 2 line 28: "summarize **our** main results"

35. Page 4 line 28: "**The** horizontal temperature field"

36. Page 7 line 16: "demonstrates typical winter behavior" (remove 'a' and 'for')

37. Page 11 line 1: "turbul**ent** energy dissipation rates"

38. Page 11 line 5: "as **the** method's uncertainty."

39. Page 11 line 10: "does not show **a** big difference"

40. Page 11 line 14: "we show **the** wavelet spectrogram of **the** neutral density fluctuations"
Interactive
comment

41. Page 12 line 6: "when **the** vertical wavelength of **the** GW become shorter"

42. Fig. 13 caption line 2: "measured during desc**ent** of WADIS-2"

43. Page 15 line 3: "spectra from **the** region just below **the** one described above **(shown in Fig. 13b)**"

44. Page 15 line 12: "They showed that this signature was observed by **the** MLS instrument over **a** large region. . . for **a** long time."

45. Page 15 line 14: "could be qualified as **a** mesospheric inversion layer"

46. Page 15 line 30: "In **the** pure adiabatic limit case"

47. Page 16 line 10: "waves with a vertical wavelength of  2-3 km **are** saturated . . . and **break**, producing turbulence layers"

48. Page 16 line 14: "the altitude region below  82km reveal**s** features similar to **those** observed. . ."

49. Page 16 line 26: "a power increase in **the** spectrum. . ."

50. Page 16 line 29: "that is **they** do not influence the flow and are conservative, i.e. **their** value is not affected by the flow"

51. Page 17 line 3: "Recalling **the** above discussion about **the** relationship. . ."

52. Page 17 line 12: "raises the question **of** how. . ."

53. Page 17 line 16: "Both spectra in the middle. . ." (remove '**the**')

54. Page 17 line 26: "The difference in their response to turbulence arise**s** due to difference of their diffusivity. Thus, **the** diffusion constant. . ."

55. Page 17 line 33: "Such models implement **the** theory of Batchelor (1958)..."

---

## Author Comment (AC1) · 18 May 2019

**Response to reviewers' reports on the paper acp-2018-1043 Simultaneous in-situ measurements of small-scale structures in neutral, plasma, and atomic oxygen densities during WADIS sounding rocket project**

Boris Strelnikov[1], Martin Eberhart[4], Martin Friedrich[5], Jonas Hedin[3], Mikhail Khaplanov[3†], Gerd Baumgarten[1], Bifford P. Williams[6], Tristan Staszak[1], Heiner Asmus[1], Irina Strelnikova[1], Ralph Latteck[1], Mykhaylo Grygalashvyly[1], Franz-Josef Lübken[1], Josef Höffner[1], Raimund Wörl[1], Jörg Gumbel[3], Stefan Löhle[4], Stefanos Fasoulas[4], Markus Rapp[2], Aroh Barjatya[7], Michael J. Taylor[8], and Pierre-Dominique Pautet[8]

[1]Leibniz-Institute of Atmospheric Physics at the Rostock University, Kühlungsborn, Germany
[2]Deutsches Zentrum für Luft- und Raumfahrt, Institut für Physik der Atmosphäre, Oberpfaffenhofen, Germany
[3]Department of Meteorology (MISU), Stockholm University, Stockholm, Sweden
[4]University of Stuttgart, Institute of Space Systems, Stuttgart, Germany
[5]Graz University of Technology, Graz, Austria
[6]GATS, Boulder, USA
[7]Embry-Riddle Aeronautical University, FL, USA
[8]Center for Atmospheric and Space Sciences, Utah State University, Logan, Utah, USA
[†]Deceased
**Correspondence:** B. Strelnikov (strelnikov@iap-kborn.de)

We appreciate the reviewers' constructive comments and their positive judgment on our paper. We have taken the reviewers' suggestions into account when preparing the revised version of our manuscript.

In the following we address the comments of both reviewers point by point

(Every point contains: **Q:** reviewer's question, **A:** authors' answer, **T:** text from the revised manuscript).

**5  To reviewer #1**

1. **Q:** *Page 1, lines 12-13: 'GW might dissipate and thereby generate turbulence'-GW might break and generates turbulence. This statement can be supported by appropriate references of observational (e.g., https://doi.org/10.1002/2015JD024283) work(s) since a lot of studies are available but nothing cited.*

   **A:** As suggested by the reviewer, we added some references to observational works showing that breaking GW may generates turbulence.

   **T:** When propagating, GW might dissipate and thereby generate turbulence (e.g., Yamada et al., 2001; Selvaraj et al., and references therein).

2. **Q:** *Page 1, lines 13-14: 'Turbulence mixing and redistribution of trace constituents' can be supported by appropriate references of observational work(s).*

   **A:** We added some references were turbulence mixing and redistribution of trace constituents were studied experimentally

   **T:** Apart of the momentum deposition, which is a key coupling process, this also affects mixing and redistribution of trace constituents (e.g., Fukao et al., 1994; Bishop et al., 2004).

3. **Q:** *Page 2, lines 6-7: 'chemical heating rates in the mesopause region of several K/day which is comparable (or even competitive) to those of turbulent heating' can be supported by appropriate references of observational work(s) whos study concluded it statistically and/or mention case study.*

   **A:** As suggested, we supported the statement by references.

   **T:** These reactions yield chemical heating rates in the mesopause region of several K/day which is comparable (or even competitive) to those of turbulent heating (Mlynczak, 1996; Lübken, 1997; Lübken et al., 2002) as well as direct heating due to solar radiation (e.g., Fomichev et al., 2004; Feofilov and Kutepov, 2012; Lübken et al., 2013, and references therein).

4. **Q:** *Page 4, lines 23-25: It is stated in Page 3, lines 13-14 that 'MAARSY operated continuously to detect PMWE echoes if any' but no echo is detected during the night of rocket launch. It can be mentioned that echo occurrence depends on both nature of the target and sensitivity of the radar. It is better to reveal with help of volume reflectivity map (without any signal threshold) during that night of rocket launch. And select the profile of volume reflectivity to discuss the sensitivity of MAARSY at that time of sounding in MLT since this article deals with small scale dynamical fluctuations in densities while avoiding chemical reactions. It can be mentioned that during February-March 2011/2012 and 2012/2013, PMWE of MAARSY appeared throughout day and night with dis-continuities in seasonal- local time variation (see Fig. 3 of Latteck and Strelnikova, 2015).*

   **A:** As requested by the reviewer to better demonstrate the dynamic conditions for WADIS-2 rocket launch we added the plot of volume reflectivity measurements by MAARSY (see Fig. 2 below and in revised version of the manuscript). We also noted that the WADIS sounding rocket mission did not aim at studying radar echoes themselves, so that the presence of PMWE was not a criterion for rocket launch.

   **T:** Fig. 2 shows volume reflectivity measured by MAARSY during 5 of March, i.e. the day of the rocket launch. Some short-living echoes were observed around noon and in the late evening, but not in the morning when WADIS-2 rocket was launched. We recall here that this sounding rocket mission did not aim at studying PMWE, so that the presence of PMWE was not a criterion for rocket launch.

[Figure]

**Figure 2.** Volume reflectivity measured by MAARSY on 5 of March. Some short-living echoes were observed around noon and in the late evening but not around the WADIS-2 rocket launch.

5. **Q:** *Page 4, line 27: 'temperature field measured by the lidars'. Is it measured or estimated? If temperature is estimated then provide appropriate references to methodology for temperature retrieval which is used here*

    **A:** The reviewer is right that it was not explained in the paper how the temperature were measured by the lidars. We believe that lidars do measure temperatures, even though this may be considered as indirect measurements. Since we show measurement results of three different lidars, to address this reviewer's point we extended the section 2 (instrumentation) by an appropriate instrument descriptions and references.

    **T:** All three lidars measure temperatures profiles along the beam direction as shown in Fig. 1. The instruments and temperature retrieval techniques for these lidars are described elsewhere (von Zahn et al., 2000; Hauchecorne and Chanin, 1980; She et al.; Arnold and She, 2003; Höffner and Fricke-Begemann, 2005; Lautenbach and Höffner, 2004; Höffner and Lautenbach, 2009). Additionally, RMR- and Na-lidars measure line-of-sight wind speed in the altitude ranges 20–80 and 80–110 km, respectively (Baumgarten, 2010; Arnold and She, 2003).

6. **Q:** *Page 5: In Fig. 2, 'temperature of RMR- and Fe- lidar are combined'. How do they agree?*

    **A:** We added a short discussion to offer reader a guideline to estimate how the RMR- and Fe-lidar measurements agree.

    **T:** Fig. 3 utilizes measurements by the vertical beam of RMR-lidar since the mobile Fe-lidar only measures vertically. Also, the seeding temperature for derivation of RMR-temperatures was taken from Fe-lidar measurements. Signatures of long period waves are clearly seen above ∼65 km altitude in both RMR- and Fe-lidar measurements.

7. **Q:** *Page 5: In Fig. 2 and Fig. 3, It is better to limit the steps in colormap for easy reference of temperature in Fig. 2 and Fig. 3. And also provide temperature labels in colormap of Fig. 3. And show the results Fig.2 and Fig. without interpolation.*

    **A:** We changed the figures 2 and 3 as requested by the reviewer (Since we added an additional plot with radar measurements above, the numbering is increased by 1, i.e. see Fig. 3 & 4). The both figures do not contain any interpolation.

**T:** See Fig. 3 and Fig. 4.

[Figure]

**Figure 3.** Combined RMR- and Fe-lidar temperature measurements during the night of the WADIS-2 rocket launch, i.e. 4 to 5 of March 2015.

[Figure]

**Figure 4.** NS and WE keogram summary of the AMTM temperature measurements obtained during the night of 4 to 5 March 2015.

8. **Q:** *Page 7: In Fig. 5, Rocket observed densities are seem to be very much smoothed. How it is smoothed?*

    **A:** see next point

9. **Q:** *Page 7: In Fig. 5, Do you find any spin frequency in up- and down-leg of rocket observations? If yes, what are the methods are used to remove those frequency and what are they?*

    **A:** We added a description of how smoothing and spin filtering was applied to the rocket-borne density measurements.

    **T:** The spin frequency of WADIS-2 rocket of 3.27 Hz which modulated the raw data was filtered out by applying a notch filter. Additionally, the shown in-situ measured densities were smoothed by running average of a length of $\sim$200 m.

10. **Q:** *Page 7: Line 14, 'Both up- and down-leg profiles are very similar in terms of oscillations'. It is worth to show the detrended densities or density-normalised density fluctuations in subplot along with Fig. 5. item Page 7: Line 17, 'some GW-signatures'. It is better to quantify the GW amplitudes corresponding to Fig. 6, as you like, below 80 km, 83-90 km and 95-100 km.*

    **A:** Unfortunately, we do not really understand what reviewer means with the *"density-normalized density fluctuations"*. Our logical structure of the paper was as follows: first to described measurements as they were and then to make some analysis, in particular, extract fluctuations. According to this logic we show the relative density fluctuations in Fig. 10 (Fig. 11 in the revised version) in the section 4 "Analysis". As we understand, these relative fluctuations, i.e. the quantities $\Delta\rho/\overline{\rho} = (\rho - \overline{\rho})/\overline{\rho}$ (where $\rho$ is measured density and $\overline{\rho}$ is a smoothed background density) are the *"density-normalized density fluctuations"* requested by the reviewer.
    The GW amplitudes are described as suggested.

    **T:** The temperature profiles clearly show some GW-signatures with amplitudes of up to 15 K at altitudes below 80 km. The height range between $\sim$83 and 90 km reveals very low GW amplitudes, i.e. temperature fluctuations of 1 K and less. A temperature increase of $\sim$40 K reminiscent of mesospheric inversion layers (MIL) similar to those analyzed by Szewczyk et al. (2013) is seen between 95 and 100 km and is discussed in Sec. 4. Some small-scale GWs with amplitudes between 1 and 5 K and vertical wavelength of the order kilometer are superposed on this large temperature disturbance.

11. **Q:** *Page 8: In Fig. 6, It is worth to describe the mothod(s) and along with valid assumptions used for temperature retrieval from densities. Include the profiles of temperature from Lidars in Fig. 6 for ready comparison of temperature osciallations between remote sensing and in-situ measurements.*

    **A:** We extended the instrumentation section with a description of derivation methods and underlying assumptions. Fig. 6 (Fig. 7 in the revised version) was changed and now includes profiles of lidar measurements.

    **T:** The measured density profile, in turn, can be integrated assuming hydrostatic equilibrium to yield temperature profile (see e.g., Strelnikov et al., 2013, for details).
    Fig. 7 additionally shows temperature profiles in magenta, yellow, and red measured by RMR-, Na-, and Fe-lidars, respectively.

[Figure]

**Figure 7.** Temperatures derived from the densities shown in Fig. 6 assuming hydrostatic equilibrium.

12. **Q:** *Page 9: Line 1, 'some aurora was seen'. It is better to mention the wave length of those observed emissions.*
    **A:** We extended the section 2 "Instrumentation" by a description of this photometer.

    **T:** The second MISU photometer on the payload measured the emission from the (0-0) band of the $N_2^+$ 1st negative band system centered at 391.4 nm. This emission is a sign of precipitating auroral electrons and thus a sensitive indicator of auroral activity.

13. **Q:** *Page 9: Line 14, 'O-density profiles reveal some oscillations'. It is better to show profiles of density-normalised density fluctuations in subpot along with Fig. 8.*
    **A:** see point 10 above.

14. **Q:** *Page 11: Fig. 10, It is better to have percentage of amplitude and densitynormalised density fluctuations in Fig. 10.*
    **A:** We are convinced that wind fluctuations in absolute values (m/s) are more informative than those in percent. However, we put here the requested figure to show the reviewer how it looks in percent.
    See also point 10 above.

[Figure]

**Figure 10.** Horizontal wind fluctuations in percent.

15. **Q:** *Page 11: Line 9, Before comparison of results, describe the observed results of turbulence intensity from different techniques (Heisenberg and Tatarskii) to demostrate the need of different techniques here. And also present the mean of them in order to present quatification of turbulence intensity.*

   **A:** The turbulence analysis technique utilized in this study is the same regardless of spectral model applied. The differences in derived energy dissipation rates arise due to numerous factors. A comparison of these differences and discussion of reasons and uncertainties is currently a subject of another paper we are working on and definitely lies out of the scope of this article. We decided to present both results (Heisenberg and Tatarskii) separately to thereby demonstrate and to allow the reader to get some filling of method uncertainties. The discrepancies of the $\varepsilon$-values derived from those two models increase when we derive mean values over extended altitude region. The presence of both Heisenberg and Tatarskii results separately, as well as the mean values over ten kilometers are shown for compatibility reasons, i.e. to enable their comparison with those used in other studies (e.g., Strelnikov et al., 2017; Szewczyk, 2015).

16. **Q:** *Page 12: Lines 6-7, 'This picture is reminiscent of a GW-saturation process when vertical wavelength of GW becomes shorter'. It is worth to quantify the GW activity using the lidar observations and compare them with Fig. 12 for quantitative and qualitative statements since this article mainly focus on dynamics of smallscales.*

   **A:** The reviewer is absolutely right that it would be valuable to make an in depth quantitative analysis of the observed gravity waves (GW) and there parameters. We are currently working on a detailed analysis of GWs observed in the course of WADIS-2 campaign and we have realized, that the detailed picture is rather complicated. In particular, we believe that we observed a superposition of many "quasi-monochromatic" GWs, that we observed many different wave packets at different altitudes, etc. We are convinced that quantification of the observed wave parameters needs a careful detailed study which is not finished yet.

   Another more in depth analysis is presented in paper by Wörl et al. (2019) which we mentioned in the manuscript.

   That is why we decided to not aim at detailed analysis of gravity waves in this paper. Rather, as we stated in the beginning "This paper aims at two things. First, it is to provide an overview of the WADIS-2 sounding rocket campaign and measured parameters, and second, to demonstrate that gravity wave motions and turbulence effects distribution of atomic oxygen in the nighttime MLT region."

17. **Q:** *Page 13: Line 14, Briefly describe the assumptions behind this Heisenberg model and the value of its constants which is used in it. How those constants are obtained?*

   **A:** We believe that by properly addressing this point we will victimize the readability of the manuscript and will loose the main focus of the paper. As we already mentioned answering the point 15 above, this point is a subject of another study that should be published in a nearest future. However, to address this referee's comment we expanded the section 2 "instrumentation" by a brief description of the turbulence derivation technique.

[Figure]

**Figure 14.** Normalized power spectral densities (PSD) of horizontal wind fluctuations (magenta), $\Delta N_n/N_n$ (black), and $\Delta[O]/[O]$ (blue) measured during descend of WADIS-2 sounding rocket. Dashed red, green, and orange lines show slopes with $k^{-3}$, $k^{-5/3}$, and $k^{-1}$ power law, respectively. Vertical black dashed and dashed-dotted lines mark inner and buoyancy scales of turbulence, $l_o$ and $L_B$, respectively. Bold gray horizontal lines mark instrumental noise. a) All spectra reveal $k^{-3}$ slope attributed to gravity waves; b) All spectra reveal both waves and turbulence. The neutral- and O-spectrum are near identical. c) Small-scale part of the O-spectrum reveal a $k^{-1}$ slope.

**T:** The derivation technique of turbulent parameters is described in detail elsewhere Lübken (1992); Lübken et al. (1993); Lübken (1997); Strelnikov et al. (2003); Strelnikov et al. (2013). Briefly, a theoretical spectral model of turbulence is fitted to a Fourier or wavelet spectrum of the measured relative density fluctuations, which are shown to be a conservative and passive turbulence tracer in MLT (Lübken, 1992; Lübken et al., 1993; Lübken, 1997). The key-feature of this technique is that the theoretical model must reproduce spectrum of turbulent tracer (scalar) in both inertial (i.e. $\propto k^{-5/3}$) and viscous (or dissipation) subranges. Transition between these subranges takes place at the so called inner scale, $l_0$, which is related to the turbulence energy dissipation rate, $\varepsilon$ as $l_0 = C(\nu^3/\varepsilon)^{(1/4)}$, where $\nu$ is the kinematic viscosity and the constant $C$ is of the order 10. Also, Lübken et al. (1993) and Lübken (1997) showed that different spectral models, in particular those by Heisenberg (1948) and by Tatarskii (1971), yield close results of the energy dissipation rates.

18. **Q:** *Page 14: Fig. 13, Fig. 13a,b & c show the spectra of neutral densities in black line. In frequencies more than 100 Hz, neutral density fluctuations are appearing as almost flattened. Is this flattening due to instrumental noise? Indicate the instrumental noise as a line in Fig. 13a,b & c?*

**A:** Yes, the almost flat part of the spectrum in Fig. 13a, b, & c (Fig. 14a, 14b, & 14c in the revised version) is due to instrumental noise. We changed the figures as suggested by the reviewer and clarified it in the text.

**T:** The spectrum of the instrumental noise is marked in Fig. 14 by a bold gray horizontal line.

19. **Q:** *Page 15: Lines 1-2, '-value is directly derived from the spatial scale lo'. Provide the formulea and appropriate references along with valid assumptions and constants.*

**A:** This comment is addressed above in scope of point 17.

20. **Q:** *Page 15: Line 15, 'MIL descended together with tide'. Quantify the tidal activity using the lidar temperature measurements since demostrated examples behave very good.*

**A:** We believe that the reviewer means that we should quantify the tidal activity in WADIS-2 measurements, whereas that sentence refers to observations by Szewczyk et al. (2013). So, we addressed this point by answering the question 21.

21. **Q:** *Page 15: Lines 18-19, 'This temperature enhancement also descends within a time period of several hours'. Provide the descend rate.*

**A:** Provided as suggested.

**T:** Our temperature enhancement (compare Fig. 7 and 3) also descends within a time period of several hours. Their temporal behavior is studied in Wörl et al. (2019), who found dominant periods of 24, 12, and 8 hours.

22. **Q:** *Page 15: Line 20, 'The upleg rocket data, however, are somewhat different'. What are the differences are observed and at what altitude?*

**A:** We changed the wordings to make it clear that the temperature enhancements on up- and downleg are similar, turbulence is only strong in the downleg data.

**T:** The upleg rocket data, however, are somewhat different. While the upleg measurements show similar temperature enhancement, the turbulence observed is one order of magnitude weaker.

23. **Q:** *Page 15: Line 22, 'MIL, it might have amplified it'. what are the sources might cause amplification of MIL at this altitude (provide appropriate references also)?*

**A:** Szewczyk et al. (2013) argued that very strong turbulence could increase temperature enhancement associated with MIL by direct frictional heating effect produced by turbulence. This statement was not a confirmed fact deduced from measurements, but rather as a suggested interpretation which was consistent with observations.

24. **Q:** *Page 15: Line 23, 'The weaker turbulence on upleg accompany ∼10 K colder temperature maximum'. I could not understand this. Can you rewrite it to make understanding in simple way.*

**A:** We rewrote the sentence to make it understandable.

**T:** The local temperature maximum near 100 km reveal ∼10 K lower values for upleg measurements than for downleg (see Fig. 7). Also turbulent energy dissipation rates measured on upleg show ∼one oder of magnitude lower values than those observed during rocket descend (Fig. 12).

25. **Q:** *Page 15: Line 26-27, 'First, the amplitudes of GW in these tracers are different. Second, the phases can be shifted relative to each other'. Can you provide values and discuss it? Since you have unique observations.*

**A:** See our answer on reviewer's comment 16.

26. **Q:** *Page 15: Lines 31-32, 'the neutral and ion density fluctuations must be in antiphase'. Is it possible to quantify the angle between neutral and ion fluctuations ($< a, b >= |a|.|b|.cos(angle)$) for those three altitudes or altitude regions*

*where coherence exist? Then, Provide the quantification of those results and discuss them.*

**A:** See our answer on reviewer's comment 16.

27. **Q:** *Page 16: Lines 6-7, 'where all three species, i.e. oxygen, ions, and neutrals show nearly the same oscillations'. It can be discussed based on quantified angles.*

    **A:** See our answer on reviewer's comment 16.

28. **Q:** *Page 17: Line 35, 'the k-1 slope at scales smaller than those where k-5/3 is present'. Provide the profiles of outer scale and inner scale as like" and indicate the altitudes in it where the spectral slope is seen as k-1. It provides the active altitude regions where two different type of diffusions take place since it is a unique measurements to deal with. And also compare these turbulence parameters with the same of low-latitude turbulence parameters since day-time low-latitude mesospheric turbulence measurements not have any affect with dusty plasma.*

    **A:** By trying to address this reviewer's comment we realized that due to very high intermittency of the observed turbulence field, marking regions with different types of spectra makes figures extremely crowded and absolutely not readable. So, we decided to comment on it in the text.

    We also believe that comparison with low-latitude turbulence parameters better fits to our newest data from recent sounding rocket campaign "PMWE" conducted in April 2018. One of the reason for that, that additionally to the rocket data we also have radar measurements, which makes it better comparable with the low-latitude measurements.

    **T:** Detailed analysis of all the regions where different types of O-density spectra occur is rather difficult because of different reasons. One reason is that when turbulence is very strong, its spectrum extends down to scales below current FIPEX resolution limit of $\sim$20 m, that is the O-density spectrum appears not fully resolved. So far we can only identify three regions where third type of spectra (like in Fig. 14c) was observed in the downleg measurements: between 90.9–91.2 km, 80.2–80.4 km, and 81.1–81.7 km.

29. Other minor corrections made as requested.

**To reviewer #2**

1. **Q:** *Page 2 line 8: provide citation for turbulent and solar heating rates.*

   **A:** Provided as requested.

   **T:** direct heating due to solar radiation (e.g., Fomichev et al., 2004; Feofilov and Kutepov, 2012; Lübken et al., 2013, and references therein).

2. **Q:** *Page 4 line 7: FIPEX - define acronym*

   **A:** Done.

   **T:** FIPEX stands for „Flux-Probe-Experiment",...

3. **Q:** *Page 4 line 9: Could you give a brief description of this technique?*

   **A:** As suggested by the reviewer, we extended the section 2 "instrumentation" by a short description of this measurement technique.

   **T:** FIPEX stands for „Flux-Probe-Experiment", it employs solid electrolyte sensors having gold electrodes with selective sensitivity towards atomic oxygen. A low voltage is applied between anode and cathode pumping oxygen ions through the electrolyte ceramic (yttria stabilized zirconia, YSZ). The current measured is proportional to the oxygen flux. A detailed description of measurements conducted by FIPEX during WADIS mission is provided by Eberhart et al. (2015) and Eberhart et al. (2018) for the first and second campaign, respectively.

4. **Q:** *Figure 2: How are the RMR and Fe lidar data combined? What is the direction of these measurements?*

   **A:** We expanded the description of the lidar measurements by addressing this comment of the reviewer.

   **T:** Fig. 3 utilizes measurements by the vertical beam of RMR-lidar since the mobile Fe-lidar only measures vertically. Also, the seeding temperature for derivation of RMR-temperatures was taken from Fe-lidar measurements.

5. **Q:** *Section 3.2, second paragraph: mention in the text (not just in the figure caption) which instrument was used to derive the total number density and temperature profiles and how the temperatures are derived from the densities.*

   **A:** To address this comment of the reviewer we extended the section 2 "instrumentation" by a short description of the absolute density and temperature derivation technique by the CONE instrument. Also, we mentioned in the text that the shown densities and temperatures were measured in-situ by the CONE instrument.

   **T:** CONE measures density of neutral air with altitude resolution of $\sim$30 cm. Making use of laboratory calibrations allows to derive absolute density altitude-profile. The measured density profile, in turn, can be integrated assuming hydrostatic equilibrium to yield temperature profile (see e.g., Strelnikov et al., 2013, for details). Figs. 6 and 7 show profiles of neutral air densities and temperatures measured in-situ by the CONE instrument.

6. **Q:** *Page 8 paragraph 2: What are the values of the ionospheric parameters used in the IMAZ model to produce the profile in Fig. 7?*

[Figure]

**Figure 8.** Densities of positive ion (solid orange and green lines) and electron (dash-dotted orange and green profiles) densities measured by the PIP and LP instruments, respectively. Bold solid red line shows results of the wave propagation experiment. Blue dashed line shows electron density from the empirical ionospheric model for the auroral zone, IMAZ, derived for the time of the WADIS-2 flight (see text for details).

**A:** As suggested by the reviewer, we added description of inputs for the IMAZ model. Additionally, we added profile of the absolute electron densities measured by the radio wave propagation experiment to the Fig. 7 (Fig. 8 in the revised version).

5     **T:** The inputs for IMAZ model are F10.7 solar flux index of 137.9 Jy (1 Jy = $10^{-26}$ W m$^{-2}$Hz$^{-1}$), the planetary magnetic Ap index 5, and riometer absorption @ 27.6 MHz of 0.076 dB. Activity indices, the solar F10.7 and Ap index, were obtained from the GSFC/SPDF OMNIWeb interface at https://omniweb.gsfc.nasa.gov. The integral riometer absorption was estimated from the electron density measurements by wave propagation experiment based on Friedrich and Torkar (1983).

7. **Q:** *"This is in accord with the fact that some aurora was seen" - by what instruments? "auroral emission detector. . . registered some auroral emission above 100 km" - how does this instrument work, what is measured? It might be useful to add this instrument to your list of rocket instruments in Sect. 2. Could you quantify "some auroral emission"?*

5    **A:** As suggested by the reviewer, we added both brief description of this instrument in section 2 "instrumentation" and short description of the measured quantity.

**T:** Section 2: The second MISU photometer on the payload measured the emission from the (0-0) band of the $N_2^+$ 1st negative band system centered at 391.4 nm. This emission is a sign of precipitating auroral electrons and thus a sensitive indicator of auroral activity.

10    Section 3: The overhead emission seen by the second MISU photometer was varying (both increasing and decreasing at times) during the flight indicating that the auroral emission was variable in time. The auroral emission was relatively weak with peak total band radiances of 700–800 Rayleighs (around $6 \cdot 10^7$ photon $\cdot$ s$^{-1}$ str$^{-1}$ cm$^{-2}$).

8. **Q:** *Page 9, line 3: "The shown plasma density profiles yield relative density measurements and therefore they were normalised. . ." This sentence seems to be the wrong way around: do you mean the measurements yield relative quantities*

15    *that are therefore normalised to produce the profiles in Fig. 7? If so, this could also be mentioned in the caption.*
    **A:** To avoid the misunderstanding we have rephrased this paragraph.

**T:** The PIP and LP instruments yield measurements of relative densities of positive ions and electrons, respectively. The wave propagation experiment yields accurate measurements of absolute electron densities, which are used to normalize the PIP- and LP-measurements. The normalization was made at an altitude of $\sim$115 km, where quasi-neutrality condition

20    is well satisfied for ionospheric plasma (see e.g., Friedrich, 2016; Asmus et al., 2017, for more details).

9. **Q:** *Figures 5, 7, and 8: Please use consistent units: m-3 is used in Figs. 5 and 7, and cm-3 is used in Fig. 8.*
    **A:** Fig. changed as suggested.

[Figure]

**Figure 9.** Atomic oxygen densities measurements. Bold orange and green lines with error-bars show photometer measurements on up- and downleg, respectively. FIPEX downleg data is shown by blue profile with shaded area showing measurement errors. Black dotted line shows SABER measurements (Level 2A, O event 20 orbit 71729). Blue dashed line shows NRLMSISE-00 model data for the time of rocket launch.

10. **Q:** *Figures 7 and 8: In the captions, I am not sure I see what is the "same as Fig. 5", except that these are atmospheric density profiles, but they are from different species, measured by different instruments!*

**A:** Corrected as suggested.

11. **Q:** *Figure 7: What is the quantity plotted from the IMAZ model? Is it ion or electron density?*

**A:** We improved both text and figure caption to make it clear, that we show electron density profile derived from the IMAZ model.

**T:** Comparison with the electron density profile from the empirical ionospheric model for the auroral zone, IMAZ (McKinnell and Friedrich, 2007) shown in Fig. 8 (blue dashed line) shows that ionization level of the ionosphere was moderately high. This is in accord with the fact that some aurora was seen throughout the night of these observations.

**Fig. caption:** Blue dashed line shows electron density from the empirical ionospheric model for the auroral zone, IMAZ, derived for the time of the WADIS-2 flight (see text for details).

12. **Q:** *Page 9 paragraph 2: I think you need to provide more details on the FIPEX instrument (either here or in Sect. 2). Could you specify why you are "mostly confident" in the descending data but not the ascending? Is this through comparison with data from the MISU photometers? Do you have an explanation as to what might be causing FIPEX to give erroneous measurements during one phase of the rocket flight but not another?*

**A:** To address this reviewer's comment we added a paragraph with brief explanation. We do not want to go in more details since they are discussed in the companion paper by Eberhart et al. (2018) in this special issue. The short story is that the FIPEX is sensitive to incoming flow because its current ultimately depends on incoming flux of the atomic oxygen. In combination with the limited time constant of the FIPEX it can result in that under unfavorable (rapidly varying) aerodynamic conditions (i.e. rapidly changing influx of O) FIPEX may somewhat underestimate the absolute value of the ambient O-density.

**T:** We note here that the FIPEX is a new instrument which was first applied for oxygen density measurements on sounding rockets during WADIS-1 rocket campaign (Eberhart et al., 2015). It showed good results and demonstrated principal possibility of a high resolution O-density measurements in MLT. WADIS-2 rocket was additionally equipped with the MISU airglow photometer that indirectly measures O-density from emission of the molecular oxygen Atmospheric Band. Such type of measurements is widely used also from the ground- or satellite-based platforms and is commonly accepted as reliable. The WADIS-2 sounding rocket comprised the both measurement techniques on the same platform which made it possible to closely compare their results. The both WADIS payloads (i.e. for the first and second campaigns) were equipped with several FIPEX sensors which were mounted at different angles relative to rocket velocity (or rocket symmetry) axis. Such multiple configuration aimed at finding the most favorable aerodynamic orientation for the FIPEX sensors. We also note that because of the supersonic rocket velocity the measurement results of most instruments on board sounding rockets require an aerodynamic correction (Gumbel et al., 1999; Gumbel, 2001; Rapp et al., 2001; Hedin et al., 2007; Staszak et al., 2015). By analyzing measurement conditions and comparing the measurement results we

chose the best quality FIPEX data for further analysis. For the detailed discussion of all FIPEX measurements the reader is referred to the companion paper by Eberhart et al. (2018).

13. **Q:** *Page 9 line 12: Are you linking the increase in oxygen density seen by SABER above 100 km to auroral activity because of the auroral emission measured at these altitudes by the rocket-bourne MISU instrument? If so, why is a similar increase not seen in the FIPEX oxygen measurements taken at the same location as those from the MISU instrument?*

**A:** The reviewer is absolutely right that we link the increase in oxygen density seen by SABER above 100 km to auroral activity only based on the MISU auroral photometer observations. We added a sentence to note that the measurement principle of the FIPEX is based on direct sensitivity to the ambient O-concentration. Also, by addressing the reviewer's comments 2 and 3 we added a brief description of the FIPEX measurement technique in section 2. All those additions will make it clear that the FIPEX is insensitive to any emissions.

**T:** We recall that the measurement principle of the FIPEX instrument is only sensitive to ambient O-density and does not react on emissions.

14. **Q:** *Page 9 paragraph 3: Again, a little more information on what the "other FIPEX sensors" measure would be helpful!*

**A:** We believe that by addressing the point 12 of the reviewer we also clarified (answered) this comment.

15. **Q:** *Page 10 line 6: "small-scale stuff": colloquialism, please rephrase*

**A:** Changed as suggested.

**T:** For the small-scale structures in MLT and, especially those produced by turbulence,...

16. **Q:** *Page 10 paragraph 3: please specify which data are used to determine the fluctuations in Fig. 10; are these up or down leg measurements?*

**A:** We clarified this point in both text and figure caption.

**T:** These fluctuations were derived from the density profiles measured on the downleg of WADIS-2 rocket flight.

[Figure]

**Figure 10.** Relative density fluctuations (=residuals) of neutral air, positive ions, and atomic oxygen shown by blue, orange, and green profiles, respectively. These fluctuations were derived from the density profiles measured on the downleg of WADIS-2 rocket flight by subtracting a running average over 5 km long vertical window.

17. **Q:** *Fig. 11: provide the references for the turbulence climatologies in the text (not just the figure legend)*

    **A:** Provided as suggested.

    **T:** The black bold and grey profiles show climatologies, that is mean seasonal values for winter (Lübken, 1997) and summer (Lübken et al., 2002), respectively.

18. **Q:** *Page 11 line 7: "Also the upleg and downleg turbulence data qualitatively agree with each other." they do not agree below 70 km.*

    **A:** We agree with the reviewer, that downleg measurements below 70 km are different. We discussed this difference further below in the text. However, we accept that to avoid confusion in has to be clearly stated already at this point.

    **T:** Also the upleg and downleg turbulence data qualitatively agree with each other except small region observed between 63 and 67 km on downleg.

19. **Q:** *Page 11 line 9: "If compared with results of our previous rocket campaign WADIS-1 that was conducted in summer (Strelnikov et al., 2017), the observed winter turbulence field does not show big difference between up- and downleg measurements." It would be helpful to expand a little here and give (maybe just a sentence) on the WADIS-1 turbulence field.*

    **A:** As suggested by the reviewer, we added a short summary on turbulence variability as found during WADIS-1 summer campaign.

    **T:** Turbulence filed in summer MLT observed during WADIS-1 campaign showed large oscillations in both space and time so that even mean $\varepsilon$-values of up- and downleg rocket measurements differed by order of magnitude (Strelnikov et al., 2017). Turbulence variability in time was studied by analyzing MAARSY and EISCAT (European Incoherent SCATter Scientific Association) radar measurements which were properly scaled based on in-situ data. The MAARSY is only capable of measuring MLT parameters if some radar echo occurs, that is it needs presence of PMSE or PMWE. PMSE occurrence rate as observed by MAARSY is close to 100 % (Latteck and Bremer, 2017) which makes it easy to study MLT in summer season. The winter echoes, PMWE, are much more rare, i.e. their occurrence rate is at most 30 % (Latteck and Strelnikova, 2015).

20. **Q:** *Page 12 line 2: define acronym PSD, power spectral density*

    **A:** Defined as suggested.

    **T:** (power spectral densities, PSD, vs frequency)

21. **Q:** *Page 12 line 6: "This picture is reminiscent of a GW-saturation process when vertical wavelength of GW becomes shorter." – could you provide a citation here?*

    **A:** Reference provided as suggested.

    **T:** (see, e.g., Fig. 5.3 in Nappo, 2002)

22. **Q:** *Page 12 paragraph 1: how does the wavelet spectrum shown in Fig. 12 for the down leg compare to that of the up leg?*

    **A:** We noted in the text that the upleg spectra qualitatively agree, that is they show similar intermittent turbulence field.

    **T:** We note that qualitatively similar picture in terms of turbulent structures could be inferred by analyzing the upleg data which is not shown here.

23. **Q:** *Page 12 line 9: are the O densities used here from the FIPEX instrument?*

    **A:** We made it now clear in the text that the spectrum of neutral air density fluctuations is derived from the measurements by CONE instrument, whereas O-density fluctuations used to derive wavelet spectrum we obtained from the FIPEX measurements.

    **T:** In Fig. 13 we further compare the neutral density spectrum derived from the CONE measurements with the spectrum of atomic oxygen density fluctuations derived from the FIPEX measurements.

24. **Q:** *Figure 13: would it be possible to highlight the altitudes of each 2D slice forming the three panels? These could also perhaps be shown in Fig. 12.*

    **A:** By trying to address this reviewer's comment we realized that due to very high intermittency of the observed turbulence field, marking regions with different types of spectra makes figures extremely crowded and absolutely not readable. So, we decided to comment on it in the text.

    **T:** Detailed analysis of all the regions where different types of O-density spectra occur is rather difficult because of different reasons. One reason is that when turbulence is very strong, its spectrum extends down to scales below current FIPEX resolution limit of $\sim$20 m, that is the O-density spectrum appears not fully resolved. So far we can only identify three regions where third type of spectra (like in Fig. 14c) was observed in the downleg measurements: between 90.9–91.2 km, 80.2–80.4 km, and 81.1–81.7 km.

25. **Q:** *Page 16 line 25: "usually attributed to GW" – could add a citation here*

    **A:** We added the citation as suggested by the reviewer.

[revised manuscript text omitted]

Žagar, N., Jelić, D., Blaauw, M., and Bechtold, P.: Energy Spectra and Inertia-Gravity Waves in Global Analyses, Journal of Atmospheric

20  Sciences, 74, 2447–2466, https://doi.org/10.1175/JAS-D-16-0341.1, 2017.

von Zahn, U., von Cossart, G., Fiedler, J., Fricke, K. H., Nelke, G., Baumgarten, G., Rees, D., Hauchecorne, A., and Adolfsen, K.: The ALOMAR Rayleigh/Mie/Raman lidar: objectives, configuration, and performance, Ann. Geophys., 18, 815–833, https://doi.org/10.1007/s005850000210, 2000.

Weinstock, J.: Saturated and Unsaturated Spectra of Gravity Waves and Scale-Dependent Diffusion., Journal of Atmospheric Sciences, 47,

25  2211–2226, https://doi.org/10.1175/1520-0469(1990)047<2211:SAUSOG>2.0.CO;2, 1990.

Wörl, R., Strelnikov, B., Viehl, T. P., Höffner, J., Pautet, P.-D., Taylor, M. J., Zhao, Y., and Lübken, F.-J.: Thermal structure of the mesopause region during the WADIS-2 rocket campaign, Atmospheric Chemistry and Physics, 19, 77–88, https://doi.org/10.5194/acp-19-77-2019, https://www.atmos-chem-phys.net/19/77/2019/, 2019.

Yamada, Y., Fukunishi, H., Nakamura, T., and Tsuda, T.: Breaking of small-scale gravity wave and transition to turbulence observed in OH airglow, Geophysical Research Letters, 28, 2153–2156, https://doi.org/10.1029/2000GL011945, 2001.

---

## Referee Report (RR1)

**Reviewer Report**

Strelnikov et al. are appreciated for their careful responses to my comments. Revised manuscript shows the reasonable improvement on methodology and presentation of results but general aspect of science is still in need of improvement especially main objective of this study. Some of my concerns are still needs to be satisfied. This article can be published after this revision if listed (following) questions are sufficiently answered. This manuscript is structured to investigate the small scale structures such as gravity waves and turbulence in neutral, plasma and atomic oxygen densities during WADIS sounding rocket project. This study deals mainly on dynamics but not much attention is paid on chemical processes which also can have control on observed atomic oxygen density fluctuations since its life time may be shorter (below 90 km) in the regions of 63-87 km where most of the observed populations of turbulence measurements are exist. So justifications are need to be made to project that atomic density fluctuations are due to turbulence. I would like to draw the attention on one of my this general question which was already asked but it was not answered sufficiently well in the revised manuscript too.

These are general questions but I can also try to pointout the potential pages and lines where those are need to be improved.

1. Page 2, lines 14-24 and Page 21, lines 18-21: It is stated in introduction that life time of atomic oxygen varies from seconds ( 50 km) to month ( 100 km). O-density mainly varies by dynamics of MLT above 90 km due to its long life time. This long life time can allow us to distinguish the generation mechanism of fluctuations. Authors concluded that observed O-density fluctuations are due to turbulence since life time of atomic oxygen is long, this may be true above 90 km but certainly not below 90 km. It is also can be mentioned that generalized conditions can vary from individual cases, those need to be justified with help of temporal variation of life time of atomic oxygen from theoretical model and observations with appropriate references in the height range of 63-87 km where most of the observed populations of turbulence measurements are exist (Fig. 12).

2. Page 7, Fig. 9 and page 12, lines 1-8: Since presented atomic oxygen densities are showing three different profiles (below 87 km in Fig. 9) from three different methods. It is true that those are not common volume observations but difference in magnitude is very large. Whether these differences arise due to chemical reactions or dynamical or methods of measurements are in question, may be supported with appropriate references for presented differences.

3. page 18, Discussion: Provide the profiles of life-time of atomic-oxygen, characteristics time scale of turbulence along with sampling time as a function of height from 60-90 km and then discuss the characteristics time scale of measured turbulence eddy in relation to the life time of atomic oxygen while accounting the time sampling of O-density measurements, to justify that measurements hold information of turbulence in measured atomic-oxygen densities.

---

## Author Response (AR2)

**Response to reviewer's report on the paper acp-2018-1043 Simultaneous in-situ measurements of small-scale structures in neutral, plasma, and atomic oxygen densities during WADIS sounding rocket project**

Boris Strelnikov[1], Martin Eberhart[4], Martin Friedrich[5], Jonas Hedin[3], Mikhail Khaplanov[3†], Gerd Baumgarten[1], Bifford P. Williams[6], Tristan Staszak[1], Heiner Asmus[1], Irina Strelnikova[1], Ralph Latteck[1], Mykhaylo Grygalashvyly[1], Franz-Josef Lübken[1], Josef Höffner[1], Raimund Wörl[1], Jörg Gumbel[3], Stefan Löhle[4], Stefanos Fasoulas[4], Markus Rapp[2], Aroh Barjatya[7], Michael J. Taylor[8], and Pierre-Dominique Pautet[8]

[1]Leibniz-Institute of Atmospheric Physics at the Rostock University, Kühlungsborn, Germany
[2]Deutsches Zentrum für Luft- und Raumfahrt, Institut für Physik der Atmosphäre, Oberpfaffenhofen, Germany
[3]Department of Meteorology (MISU), Stockholm University, Stockholm, Sweden
[4]University of Stuttgart, Institute of Space Systems, Stuttgart, Germany
[5]Graz University of Technology, Graz, Austria
[6]GATS, Boulder, USA
[7]Embry-Riddle Aeronautical University, FL, USA
[8]Center for Atmospheric and Space Sciences, Utah State University, Logan, Utah, USA
[†]Deceased

**Correspondence:** B. Strelnikov (strelnikov@iap-kborn.de)

We appreciate the reviewer's constructive comments and his/her positive judgment on our paper. We have taken all reviewer's suggestions into account when preparing the revised version of our manuscript.

In the following we address the comments of the reviewer1 point by point

**5 1st pont**

To address the 1st point we noted in the introduction that in context of present study local conditions can vary significantly and a general description may not be appropriate.

*Such generalized conditions, however, can considerably vary from individual cases and are not suitable for case studies. ... Also, a suitable justification of chemical life time of atomic oxygen as well as characteristic times for dynamical processes* 10 *have to be assessed for proper interpretation of measurements.*

**1 2nd point**

To address the point regarding large differences in O-density profiles below ∼87 km we added the following paragraph.

*Fig. 9 also reveals that below ∼85 km the differences between O-densities measured by different techniques are very large. This is a manifestation of disadvantages of the emission methods used for atomic oxygen retrieval. Thus e.g., the technique for deriving O from OH\* emission (SABER) is based on assumption of chemical equilibrium for ozone. Several recent works showed that below ∼80–87 km this assumption fails, in particular at high latitudes in March (Fig. 1 in Belikovich et al.,* 5 *2018; Kulikov et al., 2018). Deviation from chemical equilibrium for ozone results in underestimation of atomic oxygen density (Kulikov et al., 2019). Also the photometric methods of atomic oxygen retrieval via atmospheric band observation (762 nm) are not free of disadvantages. The fitting coefficients for this method were calculated at peak of atmospheric band emission (i.e., above ∼90 km) with an assumption that atomic oxygen recombination is a principle source of $O_2$ ($b^1\Sigma_g^+$). Below this altitude, and in particular around peak of OH\* layer, typically located at ∼80-88 km (e.g., Grygalashvyly et al., 2014), some additional* 10 *mechanisms of $O_2$ ($b^1\Sigma_g^+$) population may occur (Witt et al., 1979; Kalogerakis, 2019). This leads to overestimation of atomic oxygen calculated by the fit function of McDade et al. (1986).*

**2    3rd point**

To address the last reviewer's point regarding time constants at altitudes below ∼90 km we made relevant calculations and extended the discussion section by the following content.

[revised manuscript text omitted]

---

## Author Response (AR3)

**Response to reviewer's report on the paper acp-2018-1043**
**"Simultaneous in-situ measurements of small-scale structures in neutral, plasma, and atomic oxygen densities during WADIS sounding rocket project"**

by Strelnikov et al.

July 24, 2019

Dear Prof. Engel, as suggested we added the explanation of what the Kolmogorov time scale represents to the discussion of time scales:

[revised manuscript text omitted]